# *TaDiCodec*: Text-aware Diffusion Speech Tokenizer for Speech Language Modeling

**Yuancheng Wang, Dekun Chen, Xueyao Zhang, Junan Zhang, Jiaqi Li, Zhizheng Wu**
The Chinese University of Hong Kong, Shenzhen
`yuanchengwang@link.cuhk.edu.cn, wuzhizheng@cuhk.edu.cn`

## Abstract

Speech tokenizers serve as foundational components for speech language models, yet current designs exhibit several limitations, including: 1) dependence on multi-layer residual vector quantization structures or high frame rates, 2) reliance on auxiliary pre-trained models for semantic distillation, and 3) requirements for complex two-stage training processes. In this work, we introduce the ***Text-aware Diffusion Transformer Speech Codec*** (*TaDiCodec*), a novel approach designed to overcome these challenges. TaDiCodec employs end-to-end optimization for quantization and reconstruction through a diffusion autoencoder, while integrating text guidance into the diffusion decoder to enhance reconstruction quality and achieve optimal compression. TaDiCodec achieves an extremely low frame rate of **6.25 Hz** and a corresponding bitrate of **0.0875 kbps** with a **single-layer codebook** for 24 kHz speech, while maintaining superior performance on critical speech generation evaluation metrics such as Word Error Rate (WER), speaker similarity (SIM), and speech quality (UTMOS). Notably, TaDiCodec employs a single-stage, end-to-end training paradigm, and obviating the need for auxiliary pre-trained models. We also validate the compatibility of TaDiCodec in language model based zero-shot text-to-speech with both autoregressive modeling and masked generative modeling, demonstrating its effectiveness and efficiency for speech language modeling, as well as a significantly small *reconstruction-generation gap*. We will open source our code and model checkpoints. Audio samples are are available at `https:/tadicodec.github.io/`. We release code and model checkpoints at `https://github.com/AmphionTeam/TaDiCodec`.

## 1 Introduction

Recent advances have been made in both large language model (LLM)-based text-to-speech (TTS) systems [1, 2, 3, 4, 5, 6, 7, 8, 9] and spoken language models [10, 11, 12, 13, 14, 15, 16, 17, 18]. At the core of these systems lies the speech tokenizer, which converts continuous speech signals into discrete token sequences, thereby enabling the application of textual LLM paradigms to speech modeling. Beyond this, speech tokenizers play a fundamental role in bridging the text and speech modalities, forming the basis for cross-modal learning, alignment, and generation.

However, most existing speech tokenizers are suboptimal for speech language modeling. Prior works (*e.g.*, EnCodec [19], SoundStream [20], DAC [21]) primarily target speech signal compression and transmission, relying on multi-layer residual vector quantization (RVQ) and operating at high frame rates and bitrates. Such configurations make modeling with language models challenging and inefficient. More recently, several studies [5, 6, 22, 23, 24] have explored techniques for single-layer speech tokenizers. However, these approaches still fall short in reconstruction quality compared to RVQ-based tokenizers and often maintain high token rates (typically exceeding 50 tokens per second). Moreover, they usually depend on complex loss designs and adversarial training. Additionally,

39th Conference on Neural Information Processing Systems (NeurIPS 2025).

many of these models primarily optimize for acoustic-level reconstruction, resulting in discrete representations that lack semantic richness, making them suboptimal for language model modeling and causing *reconstruction-generation gap*.

Recent studies [2, 3, 4, 10, 11, 25, 26] emphasize that effective speech tokens for language modeling should exhibit low frame rates and semantic richness, which criteria that directly shape the design of modern speech tokenizers. To achieve this, several works [10, 25, 26, 27] decompose speech into semantic and acoustic tokens by distilling features from speech self-supervised learning (SSL) models [28, 29, 30, 31]. In this framework, semantic tokens exhibit improved alignment with textual representations, thereby facilitating more effective language modeling. However, preserving reconstruction quality often requires RVQ, along with intricate loss functions, adversarial objectives, and the integration of external SSL models. An alternative line of work, including systems such as CosyVoice [3], SeedTTS [2], FireRedTTS [4], and Vevo [32], adopts a two-stage design: first quantizing SSL-derived features, then training a separate diffusion model [33, 34, 35] to reconstruct speech conditioned on these tokens. While this design enables relatively low frame rates and supports a single-layer token representation, it comes with several limitations: **1) Two-stage training:** the pipeline introduces greater architectural complexity and reduced training efficiency compared to end-to-end approaches; **2) External dependency:** it relies on pre-trained SSL or supervised models for semantic feature extraction; and **3) Struggle with extreme compression:** most systems fail to achieve ultra-low token rates (*e.g.*, fewer than 20 tokens per second), which are critical for modeling efficiency and scalability.

To address the limitations of current speech tokenizers, we propose the ***Text-aware Diffusion Transformer Speech Codec*** (*TaDiCodec*), a novel model that achieves an exceptionally low frame rate of **6.25 Hz** using a **single codebook**, corresponding to a bitrate of **0.0875 kbps** for 24 kHz speech. Despite this ultra-low rate, TaDiCodec delivers high-fidelity speech reconstruction and robust performance on downstream speech language modeling tasks. Specifically: **1)** TaDiCodec unifies quantization and reconstruction within an **end-to-end diffusion autoencoder**, removing the need for separate semantic distillation or complex adversarial objectives by relying solely on diffusion loss; **2)** it enhances reconstruction quality and compression efficiency by incorporating **text and prompt guidance** into the diffusion decoder. Our design is motivated by the increasing availability of transcriptions from automatic speech recognition (ASR) systems [36, 37], and the widespread use of paired speech-text data in generative applications. In zero-shot TTS scenarios, for instance, the target text is inherently available; in end-to-end spoken language systems, speech and text tokens are typically generated jointly [12, 13, 14, 15, 16, 17, 38].

Our experiments show that TaDiCodec achieves performance comparable to or better than existing speech tokenizers in both reconstruction and downstream speech generation, while maintaining a significantly smaller gap between reconstruction and generation. In addition, it adopts a much simpler pipeline and operates with much fewer tokens. We evaluate zero-shot TTS using TaDiCodec under both autoregressive and masked language modeling settings, achieving strong results in intelligibility, speaker similarity, speech quality, and overall training and inference efficiency. A comparison with other tokenizers is presented in Figure 1 and Table 1.

The contributions of our work are summarized as follows:

- We propose *TaDiCodec*, a novel speech tokenizer with a token rate of 6.25 Hz and a bitrate of 0.0875 kbps, based on a diffusion autoencoder that jointly performs quantization and reconstruction without adversarial training, external pretrained models for semantic distillation, or multi-stage training. This design enables efficient optimization and simplifies the speech tokenization pipeline.

- We introduce text-aware and prompt-guided decoding into the diffusion process to facilitate extreme compression. By leveraging paired speech-text data, this approach enhances reconstruction quality and enables high intelligibility, speaker similarity, and speech quality under ultra-low token rates.

- We build zero-shot TTS models using our tokenizer under both autoregressive and masked language modeling settings, achieving WERs of 2.28 and 1.19 on *SeedTTS test-en* and *test-zh*, respectively. Our models demonstrate notable improvements on challenging benchmarks such as articulatory, code-switching, and cross-lingual test sets, and support real-time inference with RTFs ranging from 0.12 to 0.29 across different model sizes.

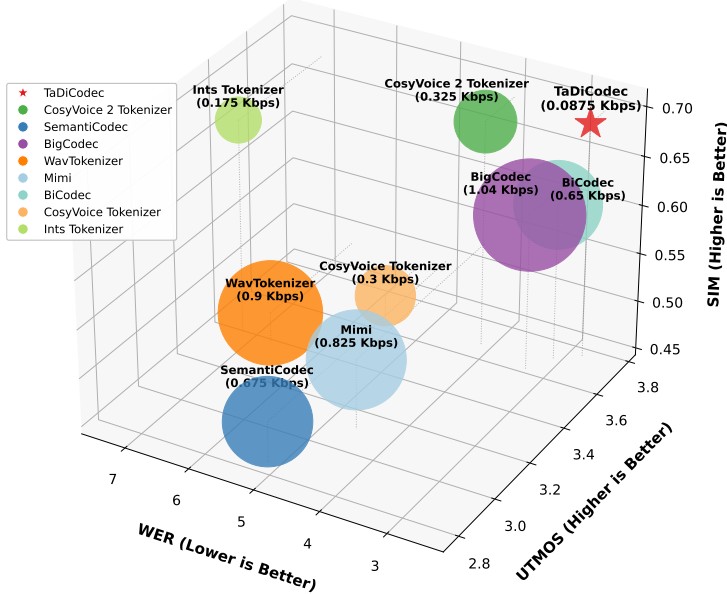

Figure 1: **Comparison between TaDiCodec and other speech tokenizers.** We use a three-dimensional coordinate system to display the performance across three dimensions: the x-axis represents WER, the y-axis represents UTMOS, and the z-axis represents SIM. The size of the markers is proportional to the kbps value.

## 2 Related Work

**Discrete Speech Tokenizer** Discrete speech tokenizers convert continuous speech into discrete tokens, enabling modern zero-shot TTS and speech language modeling. Early tokenizers [19, 20, 21] focused on audio compression, relying on residual vector quantization (RVQ) [20, 39] and operating at high frame rates and bitrates, settings ill-suited for language modeling. Recent work has shifted toward designing tokenizers tailored for language modeling, emphasizing low frame rates [10, 26], semantic-rich representations [4, 5, 6, 10, 25, 26, 27, 32, 40, 41], and single-layer codebooks [22, 23, 24]. Diffusion-based methods [33, 34] have gained popularity for their performance at low token rates and scalability. However, they typically follow a two-stage pipeline: extracting tokens via self-supervised speech representations [28, 29, 30, 31, 36, 42], then reconstructing waveforms through diffusion. For example, [43, 44] apply diffusion to improve de-tokenization quality, but still operate at relatively high token rates. Achieving ultra-low bitrates (*e.g.*, below 0.2 kbps or 20 tokens/s) with a compact, generative-friendly framework remains a major challenge.

**Zero-shot TTS** Modern zero-shot TTS systems typically operate on discrete speech tokens using either autoregressive (AR) language modeling [1, 2, 3, 4, 5, 6, 8, 9, 41] or masked generative (language) modeling (MGM) [7, 45, 46, 47]. Some models [2, 3, 4, 7, 41] adopt an "AR + diffusion" framework, where a diffusion decoder enhances waveform quality based on predicted tokens. Zero-shot TTS is also foundational in recent end-to-end spoken language models. For example, Qwen2.5-Omni [18] uses a "talker" module to generate speech tokens from the text output of a "thinker." Similar architectures [15, 16] decode speech directly from text, while others [11, 12, 14, 17] leverage TTS models to synthesize large-scale speech corpora for training dialogue agents.

## 3 Method

### 3.1 TaDiCodec

**Speech Tokenization with Diffusion Transformer Autoencoder** Some speech tokenizers adopt raw waveform signals as modeling targets. However, raw waveforms often contain a considerable amount of redundant information. In this work, we instead adopt the mel-spectrogram as both the input and reconstruction target for the tokenizer, given its compactness and ease of inversion to waveform using vocoder models [48, 49]. Formally, we denote the input mel-spectrogram as

$x \in \mathbb{R}^{T \times d}$, where $T$ denotes the number of frames, corresponding to the number of waveform frames divided by the hop size. The tokenizer's encoder $\mathcal{E}$ transforms $x$ into a sequence of latent embeddings, *i.e.*, $\mathcal{E}(x)$. These embeddings are then quantized by the vector quantization (VQ) module $Q$ into a discrete token sequence $q = Q(\mathcal{E}(x)) \in \mathbb{Z}^{T_q \times 1}$, where $T_q$ is the length of the token sequence, typically equal to $T$ divided by a predefined down-sampling factor. Each token $q_i$ (for $i \in [0, T_q)$) corresponds to an index in a codebook. The decoder $\mathcal{D}$ subsequently reconstructs the mel-spectrogram as $\hat{x} = \mathcal{D}(q)$. Previous speech tokenizers primarily adopted generative adversarial networks (GANs) [50] for training the system, typically operating on short speech segments (*e.g.*, 1–3 seconds) and employing CNNs as the backbone. However, GANs often suffer from issues related to training stability and efficiency, and the reliance on CNN-based architectures and short-segment training further constrains the model's ability to capture long-range dependencies, leading to a focus on only local acoustic patterns. To overcome these limitations, we use a fully Transformer-based [51] architecture for both the encoder and decoder, and adopt a diffusion loss for reconstruction training, enabling more stable optimization and improved modeling capabilities. Specifically, we adopt a flow matching-based decoder [35, 52]. During training, we sample Gaussian noise $\epsilon$ and generate a noisy target $x_t$ via a linear interpolation: $x_t = tx + (1 - t)\epsilon$, where $t \in [0, 1]$ is a randomly sampled noise level. The model is then trained to predict the velocity field $v$, defined as the derivative of $x_t$ with respect to $t$, *i.e.*, $v = \frac{dx_t}{dt} = x - \epsilon$. We provide more details about flow matching in Appendix B.

**Binary Spherical Quantization** For quantization, we use Binary Spherical Quantization (BSQ) [53], which does not rely on an explicit learnable codebook. We first apply downsampling to the encoder output $\mathcal{E}(x)$, followed by a linear projection to obtain a low-dimensional latent sequence: $h = \text{Linear}(\text{Downsample}(\mathcal{E}(x))) \in \mathbb{R}^{T_q \times L}$, where $T_q$ is the number of quantized frames and $L$ is the latent dimension. Each vector $h_t \in \mathbb{R}^L$ of $h$ is then projected onto the unit sphere: $u_t = \frac{h_t}{\|h_t\|}$. Binary quantization is applied independently on each dimension: $\hat{u}_t = \frac{1}{\sqrt{L}} \text{sign}(u_t)$, where $\text{sign}(x)$ is the element-wise sign function. To enable gradient flow through the quantization step, we adopt a Straight-Through Estimator (STE): $\text{sign}_{\text{STE}}(x) = \text{sg}(\text{sign}(x) - x) + x$,

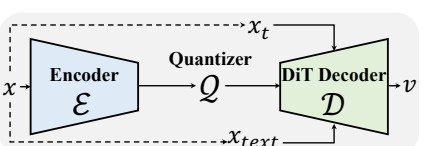

Figure 2: **Training speech tokenizer with diffusion autoencoder.** We optimize tokenization and reconstruction end-to-end with diffusion loss. The input $x$ is passed through the encoder and quantizer to get $Q(\mathcal{E}(x))$, which is then conditioned and input into the DiT decoder to predict the velocity $v$ corresponding to the noisy $x_t$.

where $\text{sg}(\cdot)$ denotes the stop-gradient operation. The quantized latent sequence $\hat{u} \in \mathbb{R}^{T_q \times L}$ is then mapped back to the $d$-dimensional space and upsampled to the original temporal resolution: $\text{Upsample}(\text{Linear}(\hat{u})) \in \mathbb{R}^{T \times d}$. Each quantized vector $h_t$ corresponds to a discrete token index computed by:

$$k_t = \sum_{i=1}^{L} 1_{[h_{t,i} > 0]} \cdot 2^{i-1}, \tag{1}$$

where $1_{[\cdot]}$ is the indicator function. As noted in [53], BSQ can be optimized without the need for a commitment loss [54], since its quantization error is theoretically bounded. This property enables end-to-end training of the system using only the diffusion loss. See Appendix C for further details.

**Text-aware De-Tokenization** Most existing speech tokenizers rely solely on speech features for reconstruction. *However, in the context of speech language modeling, the corresponding text associated with the speech is often readily available.* For example, in TTS, the target text is always known, and in most end-to-end spoken dialogue systems, text and speech tokens are generated jointly [10, 11, 12, 13, 14, 15, 16, 17, 18]. Motivated by this observation, we propose a **text-aware de-tokenization** strategy, which conditions the diffusion decoder on the corresponding text sequence $x_{text}$. To further improve reconstruction quality under the extremely low compression rate setting, we introduce a **prompt mechanism** into TaDiCodec, similar to prior works [7, 55, 56, 57]. This mechanism enables the model *to better reconstruct speech when a prompt is provided, making it particularly suitable for speech generation scenarios such as zero-shot TTS and the decoding stage of spoken language models.* Specifically, during training, we randomly sample a prefix $x_{prompt}$ from the input mel-spectrogram by drawing a segment length $l \sim \text{Uniform}(0, 0.25L)$, where $L$ denotes the total number of frames in the mel-spectrogram. The prefix is preserved without any added noise, while the loss is computed solely on the noisy portion of the sequence. Table 4 shows this prompt

mechanism yields substantial improvements in reconstruction performance. We also experiment with removing text conditioning from the decoder and observe significant performance degradation under extremely low token rate and bitrate settings. *e.g.*, at a frame rate of 12.5 Hz, the WER exceeds 10.

Notably, Unlike prior works [2, 3, 4, 7, 11, 32, 40, 58] that adopt a two-stage pipeline: first training a VQ model and then a separate diffusion model for de-tokenization, our tokenizer **jointly learns feature quantization and reconstruction in an end-to-end manner**. The overall training objective of TaDiCodec can be formulated as:

$$\mathcal{L}_{\text{diff}} = \mathbb{E}_{(\boldsymbol{x},\boldsymbol{x}_{text}),\boldsymbol{\epsilon},t} \left[ \|(\boldsymbol{x} - \boldsymbol{\epsilon}) - \mathcal{D}_\phi(Q(\mathcal{E}_\theta(\boldsymbol{x})),\boldsymbol{x}_t,t,\boldsymbol{x}_{text})\| \right], \quad (2)$$

where $\mathcal{E}_\theta$ and $\mathcal{D}_\phi$ are the encoder and decoder parameterized by $\theta$ and $\phi$. We ignore the prompt for simplification. We also find that continuing to train the decoder while freezing the encoder and VQ module can further improve performance.

## 3.2 Speech Language Modeling with TaDiCodec

Existing speech tokenizers often neglect their effectiveness in downstream speech language modeling tasks and suffer from a pronounced *reconstruction–generation gap*. In this work, we apply our tokenizer to large-scale multilingual zero-shot TTS, adopting an *"AR + Diffusion"* paradigm: an autoregressive model first predicts speech tokens $\boldsymbol{q}$ from text $\boldsymbol{x}_{text}$, which are then passed, along with the text, to TaDiCodec's diffusion decoder to generate speech. The AR model, parameterized by $\psi$, is optimized to minimize the negative log-likelihood of the target token sequence conditioned on the input text and previously predicted tokens:

$$\mathcal{L}_{\text{AR}} = -\mathbb{E}_{(\boldsymbol{q},\boldsymbol{x}_{text})} \sum_{i=1}^{T_q} \log p(\boldsymbol{q}_i \mid \boldsymbol{q}_{<i}, \boldsymbol{x}_{text}; \psi), \quad (3)$$

where $\boldsymbol{q}_i$ is the *i*-th token of $\boldsymbol{q}$. We also apply the non-autoregressive Masked Generative Modeling (MGM) [7, 59] for modeling speech tokens. See more details about MGM in the Appendix D.

## 4 Experiments

We first describe the implementation details and datasets (Section 4.1). We then present the speech reconstruction results of TaDiCodec in Section 4.2, including the main results (Section 4.2.1, Table 1), multilingual performance (Table 2), subjective evaluation results (Table 3), and ablation studies on tokenizer design (Section 4.2.2, Table 4). Section 4.3 reports the zero-shot TTS results of models built upon TaDiCodec (Table 5), along with results on model size scaling and training and inference efficiency (Table 6), and an analysis of the reconstruction–generation gap (Figure 3).

### 4.1 Experimental Settings

**Datasets** We use the Emilia [60] dataset to train all of our models. Emilia is a multilingual and diverse in-the-wild speech dataset designed for large-scale speech generation. It contains 46.8K hours of English, 49.9K hours of Chinese, 1.6K hours of German, 1.4K hours of French, 1.7K hours of Japanese, and 0.2K hours of Korean.

**Implementation Details** We build TaDiCodec using standard Llama-style Transformer blocks [61], with bidirectional attention instead of causal attention. The base configuration employs an 8-layer encoder and a 16-layer decoder, each with hidden size 1024, intermediate size 4096, and 16 attention heads. We further explore decoder variants; see Section 4.2.2 and Table 4 for details. We adopt RoPE positional embedding [62] and RMSNorm [63]. For the text-aware diffusion decoder, RMSNorm is modified to Adaptive RMSNorm to condition on the diffusion step embedding. Text tokens are adapted from a pretrained LLM vocabulary [64, 65], and concatenated with speech features along the time axis before being input to the decoder. For vector quantization, we use BSQ [53] with a latent size of 14, yielding a codebook size of $2^{14} = 16384$. All models are trained on 8 80GB NVIDIA A100 GPUs using dynamic batching with 200 seconds of speech per batch. We train the tokenizer for 800K steps using AdamW [66] with a learning rate of $7.5 \times 10^{-5}$ and 32K warmup steps. TTS models are trained for 300K steps with a learning rate of $3 \times 10^{-4}$ unless otherwise specified. AR models extend the vocabulary of pretrained textual LLMs [3, 5] and are trained with 0.2B, 0.5B, 3.0B, and 4.0B parameters; see Section 4.3 for analysis. For MGM models, we follow the setup of [7].

Table 1: **The comparison between TaDiCodec and other speech tokenizers.** TaDiCodec offers an extremely high compression rate, achieving a 6.25 Hz frame and token rate and a 0.0875 kbps bitrate without requiring additional pretrained models for semantic distillation. It achieves comparable or better reconstruction quality than other speech tokenizers, based on generation-related metrics.

| System | Frame Rate | Token Rate | Bitrate (kbps) | Codebook Number | Semantic Distill Free | Reconstruction Quality | | |
|---|---|---|---|---|---|---|---|---|
| | | | | | | WER (↓) | SIM (↑) | UTMOS (↑) |
| *Token rate less than 150* | | | | | | | | |
| EnCodec [19] | 75 | 150 | 1.5 | 2 | ✓ | 5.36 | 0.48 | 1.54 |
| DAC (RVQ) [21] | 25 | 75 | 0.75 | 3 | ✓ | 20.08 | 0.39 | 1.75 |
| DAC (VQ) [21] | 75 | 75 | 0.75 | 1 | ✓ | 12.74 | 0.45 | 2.08 |
| SpeechTokenizer [27] | 50 | 100 | 1 | 2 | ✗ | 7.98 | 0.46 | 2.47 |
| Mimi [10] | 12.5 | 75 | 0.825 | 6 | ✗ | 4.51 | 0.52 | 3.09 |
| Mimi [10] | 12.5 | 100 | 1.1 | 8 | ✗ | 3.99 | 0.57 | 3.21 |
| DualCodec [26] | 12.5 | 75 | 0.925 | 6 | ✗ | 2.63 | 0.62 | 3.78 |
| DualCodec [26] | 12.5 | 100 | 1.225 | 8 | ✗ | 2.57 | 0.64 | 3.78 |
| BiCodec [6] *16 kHz* | 50 | 50 | 0.65 | 1 | ✗ | 3.05 | 0.61 | 3.68 |
| X-codec 2 [5] *16 kHz* | 50 | 50 | 0.8 | 1 | ✗ | 2.63 | 0.62 | 3.68 |
| WavTokenizer [23] | 75 | 75 | 0.9 | 1 | ✓ | 6.65 | 0.48 | 3.36 |
| BigCodec [22] *16 kHz* | 80 | 80 | 1.04 | 1 | ✓ | 3.25 | 0.61 | 3.59 |
| TAAE [24] *16 kHz* | 25 | 25 | 0.4 | 1 | ✓ | 11.08 | 0.41 | 3.87 |
| *Two stage, Diffusion decoder* | | | | | | | | |
| SemantiCodec [40] | 25 | 50 | 0.675 | 2 | ✗ | 5.11 | 0.49 | 2.83 |
| Vevo Tokenizer [32] | 50 | 50 | 0.65 | 1 | ✗ | 3.04 | 0.53 | 3.50 |
| FireRedTTS Tokenizer [4] | 25 | 25 | 0.35 | 1 | ✗ | 3.35 | 0.59 | 3.40 |
| CosyVoice Tokenizer [3] | 25 | 25 | 0.3 | 1 | ✗ | 5.63 | 0.47 | 3.65 |
| CosyVoice 2 Tokenizer [41] | 25 | 25 | 0.325 | 1 | ✗ | 4.10 | 0.68 | 3.65 |
| *Token rate less than 20* | | | | | | | | |
| *Two stage, Diffusion decoder* | | | | | | | | |
| Ints Tokenizer [68] | 12.5 | 12.5 | 0.175 | 1 | ✗ | 7.14 | 0.67 | 3.37 |
| *One stage, Diffusion decoder* | | | | | | | | |
| **TaDiCodec** | **6.25** | **6.25** | **0.0875** | **1** | ✓ | **3.02** | **0.67** | **3.68** |
| **TaDiCodec (w. dct)\*** | **6.25** | **6.25** | **0.0875** | **1** | ✓ | **2.73** | **0.69** | **3.73** |

* "w. dct" denotes continued training of the decoder for 400K additional steps, with the encoder and VQ module frozen.

**Evaluation** We evaluate our approach from two main perspectives: speech reconstruction using the proposed tokenizer (Section 4.2) and zero-shot TTS performance (Section 4.3). We assess intelligibility (WER), speaker similarity (SIM), and speech quality (UTMOS). Speaker similarity is computed as the cosine similarity between `WavLM-TDNN` embeddings of the prompt and generated speech [28]. WER is measured using `whisper-large-v3` [36] for non-Chinese languages and `paraformer-zh` [37] for Chinese, following prior work [2, 3, 7, 58]. Speech quality is evaluated using the official UTMOS checkpoint. In addition to objective metrics, we conduct subjective evaluation via Comparative Mean Opinion Score (CMOS). We do not report signal-level metrics (*e.g.*, PESQ, STOI), as our focus is on generation-oriented performance, in line with [58, 67]. Further Evaluation details are provided in Appendix F.

## 4.2 Speech Reconstruction

### 4.2.1 Main Results

We report our main results on *SeedTTS test-en* [2] in Table 1. We also evaluate our methods on multilingual test sets in Table 2. Subjective evaluation results are show in Table 3.

**Baselines** We compare with a wide range of baselines in settings where the token rate is less than 150: **1)** single stage with multi-layer codebook and adversarial training: EnCodec [19], DAC [21], SpeechTokenizer [27], Mimi [10], DualCodec [26]; **2)** single stage with single-layer codebook and adversarial training: DAC (with single VQ), BiCodec [6], X-codec 2 [5], WavTokenizer [23], BigCodec [22], TAAE [24]; **3)** two stage with diffusion decoder: SemantiCodec [40], Vevo Tokenizer [32], FireRedTTS Tokenizer [4], CosyVoice [3] & CosyVoice 2 Tokenizer [41], Ints Tokenizer [68]. We provide more detailed description of these baselines in Appendix E.1.

**Results Analysis** **1) Compression:** TaDiCodec demonstrates a significantly higher compression rate compared to all baselines. It operates at a token rate of 6.25 Hz with a single-layer codebook, resulting in a bitrate of 0.0875 kbps. Among the baselines, the closest in compression rate to TaDiCodec is the Ints Tokenizer, which has double the token rate and bitrate of TaDiCodec. However, it performs worse in terms of WER (7.14 vs. 2.73) and UTMOS (3.37 vs. 3.73) and requires two-stage training and semantic distillation. All other baselines have a token rate greater than 25 and a bitrate of at least 0.3 kbps. Compared to other single-stage and distillation-free models, BigCodec has a higher

Table 2: **Results of multilingual speech reconstruction.** In addition to English, we evaluate on five other languages: Chinese (zh), French (fr), German (de), Japanese (ja), and Korean (ko).

| System | Bitrate | en | | zh | | fr | | de | | ja | | ko | |
|---|---|---|---|---|---|---|---|---|---|---|---|---|---|
| | (kbps) | WER | SIM | WER | SIM | WER | SIM | WER | SIM | WER | SIM | WER | SIM |
| Mimi [10] | 1.1 | 3.99 | 0.57 | 2.87 | 0.59 | 20.71 | 0.55 | 16.12 | 0.59 | 25.71 | 0.44 | 36.10 | 0.57 |
| BiCodec [6] *16 kHz* | 0.65 | 3.05 | 0.61 | 1.97 | 0.66 | 17.74 | 0.57 | 11.98 | 0.64 | 20.50 | 0.49 | 29.39 | 0.63 |
| FireRedTTS Tokenizer [4] | 0.35 | 3.35 | 0.59 | 1.99 | 0.68 | 20.16 | 0.56 | 13.87 | 0.61 | 18.57 | 0.48 | 32.20 | 0.62 |
| **TaDiCodec (w. dct)** | **0.0875** | 2.73 | 0.69 | 0.94 | 0.75 | 20.29 | 0.69 | 11.77 | 0.73 | 20.22 | 0.59 | 26.80 | 0.74 |

WER (3.25 vs. 2.73) and lower SIM (0.61 vs. 0.69) than TaDiCodec, with a bitrate of 1.04 kbps. Models with lower bitrates, such as TAAE, still have bitrates four times higher than ours and perform significantly worse in WER and SIM. Other single-layer codebook tokenizers like BiCodec, X-codec 2, and WavTokenizer have bitrates 7.4, 9.1, and 10.3 times higher, respectively. **2) Reconstruction Quality:** In terms of WER, TaDiCodec achieves a score of 3.02 without decoder continued-training and 2.73 with fine-tuning, ranking just behind DualCodec and X-codec 2, which have scores of 2.57 and 2.63, respectively, but with bitrates 10.6 and 9.1 times higher. Table 4 shows that our setting with a bitrate of 0.175 kbps achieves the best WER. In terms of SIM, TaDiCodec with decoder continued-training achieves the best SIM of 0.69, while even without decoder continued-training, it reaches 0.67, surpassing all baselines except for the CosyVoice 2 tokenizer. In terms of UTMOS, our model achieves scores of 3.68 and 3.73 (with and without decoder continued-training), ranking just behind DualCodec and TAAE, which have scores of 3.78 and 3.87. However, these models operate at much higher bitrates of 0.925 kbps and 0.4 kbps and demonstrate poorer performance in SIM.

**Results for Multilingual** As shown in Table 2, TaDiCodec achieves the best WER on English, Chinese, German, and Korean, with especially low WER on Chinese. It also outperforms all baselines in speaker similarity across all evaluated languages.

**Subject Evaluation Result** As shown in Table 3, our proposed system achieves the highest CMOS score among evaluated baselines. More details about subjective evaluation are shown in Appendix F.3.

### 4.2.2 Ablation Study

In this section, we explore several designs for TaDiCodec. For the ablation study, we report the results on *SeedTTS test-en* and *SeedTTS test-zh*. **1) Vector Quantization Scheme:** Replacing BSQ with a standard VQ tokenizer (implemented following [21, 69] with an explicit codebook of the same size as BSQ) leads to consistent degradation across all evaluation metrics. This indicates that BSQ more effectively preserves both speech quality and intelligibility. **2) Tokenizer Size Scaling:** Reducing the decoder size to 160M results in substantial performance drops, particularly in English WER. In contrast, increasing the decoder size results in marginal improvements. These results also imply the existence of a scaling law for TaDiCodec, warranting further investigation in future work. **3) Prompt Mechanism:** The introduction of the prompt mechanism substantially improves all three evaluation metrics. A plausible explanation is that the prompt serves as a global conditioning signal (*e.g.*, speaker identity), thereby reducing the quantizer's burden to encode such global information. **4) Inference Time Scaling:** Increasing the number of inference steps to 50 yields marginal improvements, while reducing it to 10 leads to slight degradation. However, further reduction to 5 steps results in a noticeable drop in performance. Considering the trade-off between efficiency and quality, using 10 to 32 steps appears to be a reasonable operating range. We aim to achieve comparable performance with fewer inference steps (*e.g.*, 1-2 steps) by leveraging techniques such as [70, 71, 72]. **5) Decoder Continued-training:** We explore freezing the encoder and the VQ module while only continued-training the decoder for an additional 400K steps, focusing solely on reconstruction. This approach yields further improvements, with WER dropping from 3.02 to 2.73 for English and from 1.11 to 0.94 for Chinese. SIM also improves for both languages. **6) Diffusion vs. GAN:** We also replace the diffusion loss with PatchGAN [73], but observe a noticeable performance drop in both intelligibility and speech quality.

### 4.3 Zero-shot TTS

In this section, we present the zero-shot TTS results using TaDiCodec as the prediction target. We evaluate two different language modeling approaches: autoregressive (AR) and masked generative

Table 3: **Subjective CMOS scores.** We randomly choose 40 samples from a in-the-wild data source. Comparisons between different models can also be found in demo page.

| System | CMOS |
|---|---|
| Ground Truth | +0.28 ±0.25 |
| Mimi [10] | -1.79 ±0.13 |
| WavTokenizer [28] | -1.33 ±0.28 |
| DualCodec [26] | -0.92 ±0.31 |
| X-codec 2 [5] | -1.07 ±0.19 |
| **TaDiCodec** | 0.00 |

Table 4: **Ablation study.**

| System | Recon. *Seed en* | | | Recon. *Seed zh* | | |
|---|---|---|---|---|---|---|
| | WER | SIM | UTMOS | WER | SIM | UTMOS |
| TaDiCodec | 3.02 | 0.67 | 3.68 | 1.11 | 0.74 | 2.70 |
| *bsq → vq* | 3.30 | 0.64 | 3.44 | 1.25 | 0.72 | 2.46 |
| *w. prompt → wo. prompt* | 8.63 | 0.52 | 3.26 | 5.42 | 0.59 | 2.28 |
| *decoder size: 320M → 160M* | 7.96 | 0.63 | 3.60 | 2.02 | 0.73 | 2.89 |
| *decoder size: 320M → 480M* | 2.90 | 0.69 | 3.68 | 1.02 | 0.75 | 2.73 |
| *frame rate: 6.25 hz → 12.5 hz* | 2.57 | 0.69 | 3.58 | 1.09 | 0.75 | 2.68 |
| *Inference steps: 50* | 2.87 | 0.68 | 3.66 | 1.07 | 0.75 | 2.68 |
| *Inference steps: 10* | 3.85 | 0.67 | 3.65 | 1.23 | 0.74 | 2.69 |
| *Inference steps: 5* | 7.89 | 0.65 | 3.19 | 1.96 | 0.73 | 2.35 |
| *w. decoder continued-training* | 2.73 | 0.69 | 3.73 | 0.94 | 0.75 | 2.69 |

Table 5: **The zero-shot TTS results.** Beyond regular cases, we also evaluate on challenging scenarios, including articulatory, code-switching, and cross-lingual settings.

| System | Frame Rate | Regular | | | | Articulatory | | | | Code-switching | | | | Cross-lingual | | | |
|---|---|---|---|---|---|---|---|---|---|---|---|---|---|---|---|---|---|
| | | en | | zh | | en | | zh | | en | | zh | | zh2en | | zh2en | |
| | | WER | SIM | WER | SIM | WER | SIM | WER | SIM | WER | SIM | WER | SIM | WER | SIM | WER | SIM |
| *Baseline systems* | | | | | | | | | | | | | | | | | |
| *NAR* | | | | | | | | | | | | | | | | | |
| MaskGCT [7] | 50 | 2.40 | 0.71 | 2.28 | 0.77 | 14.50 | 0.69 | 10.35 | 0.74 | 38.39 | 0.63 | 19.73 | 0.76 | 8.47 | 0.70 | 16.22 | 0.56 |
| F5-TTS [56] | 93.75 | 3.02 | 0.63 | 3.87 | 0.71 | 14.13 | 0.61 | 19.54 | 0.66 | 35.35 | 0.54 | 32.63 | 0.68 | 19.93 | 0.64 | 13.78 | 0.46 |
| *AR* | | | | | | | | | | | | | | | | | |
| ARS [7] | 50 | 3.55 | 0.68 | 4.37 | 0.75 | 15.98 | 0.68 | 24.07 | 0.71 | 48.59 | 0.63 | 59.71 | 0.76 | 15.22 | 0.70 | 24.30 | 0.56 |
| CosyVoice 2 [41] | 25 | 2.89 | 0.66 | 1.29 | 0.76 | 8.63 | 0.66 | 7.60 | 0.74 | 28.32 | 0.59 | 38.39 | 0.75 | 9.98 | 0.67 | 7.59 | 0.53 |
| FireRedTTS [4] | 25 | 8.53 | 0.46 | 1.27 | 0.65 | 14.47 | 0.45 | 18.81 | 0.64 | 15.03 | 0.38 | 23.97 | 0.63 | 3.87 | 0.34 | 9.04 | 0.48 |
| Ints [68] | 12.5 | 3.43 | 0.65 | 2.85 | 0.73 | 12.75 | 0.65 | 11.41 | 0.69 | 26.30 | 0.57 | 19.46 | 0.73 | 9.43 | 0.65 | 10.13 | 0.49 |
| SparkTTS [6] *16 kHz* | 50 | 2.50 | 0.57 | 1.78 | 0.66 | 10.19 | 0.57 | 13.37 | 0.65 | 15.12 | 0.46 | 16.86 | 0.65 | 9.73 | 0.58 | 4.88 | 0.40 |
| Llasa [5] *16 kHz* | 50 | 3.94 | 0.58 | 8.02 | 0.64 | 11.36 | 0.55 | 21.20 | 0.58 | 17.56 | 0.46 | 26.98 | 0.59 | 26.47 | 0.49 | 9.18 | 0.41 |
| *Ours* | | | | | | | | | | | | | | | | | |
| *NAR* | | | | | | | | | | | | | | | | | |
| TaDiCodec-MGM *25 steps* | 6.25 | 3.69 | 0.65 | 1.51 | 0.75 | 10.67 | 0.63 | 8.97 | 0.71 | 14.76 | 0.57 | 20.01 | 0.73 | 9.95 | 0.65 | 4.75 | 0.48 |
| TaDiCodec-MGM *10 steps* | 6.25 | 3.85 | 0.65 | 1.69 | 0.75 | 10.78 | 0.63 | 9.81 | 0.70 | 14.94 | 0.57 | 20.78 | 0.73 | 11.08 | 0.65 | 4.66 | 0.48 |
| *AR* | | | | | | | | | | | | | | | | | |
| **TaDiCodec-AR** | 6.25 | 2.28 | 0.65 | 1.19 | 0.75 | 8.23 | 0.63 | 8.74 | 0.70 | 9.16 | 0.57 | 16.09 | 0.73 | 7.67 | 0.64 | 2.91 | 0.48 |

modeling (MGM) and we denote our models as "TaDiCodec-AR" and "TaDiCodec-MGM" respectively. The results are reported on eight test sets, including *SeedTTS test-en* and *SeedTTS test-zh*, referred to as *Regular en* and *Regular zh*, which are widely adopted benchmarks for TTS evaluation [2, 3, 5, 6, 7, 41, 56]. In addition, we report performance on more challenging test sets, proposed in [68], covering articulatory scenarios (such as repeated words and tongue twisters), code-switching, and cross-lingual settings. We provide more details about the evaluation datasets in Appendix F.1.

**Baselines**   We compare with a wide range of open-source and state-of-the-art baselines including: **1)** AR-based Systems: ARS [7], CosyVoice 2 [41], FireRedTTS [4], Ints [68], SparkTTS [6], Llasa [5]; **2)** NAR-based systems: MaskGCT [7] and F5-TTS [56]. We provide more detailed description of these baselines in Appendix E.2.

**Main Results**   We report the main results of our models and baselines on eight test sets in Table 5. Our models exhibit significant improvements in intelligibility while maintaining speaker similarity comparable to state-of-the-art zero-shot TTS systems. In terms of WER, **TaDiCodec-AR achieves the best performance on the *Regular en* and *Regular zh* test sets, reaching 2.28 and 1.19 respectively**, and outperforming all baselines. On more challenging test sets, TaDiCodec-AR demonstrates even more pronounced advantages, for example, reducing WER from 15.03 to 9.16 on *Code-switching en*, and from 4.88 to 2.91 on *Cross-lingual en2zh*. Notably, these improvements are achieved without any task-specific optimization or reinforcement learning fine-tuning [74, 75] on WER, as done in work such as [41]. For TaDiCodec-MGM, it consistently outperforms or matches the performance of state-of-the-art NAR zero-shot TTS systems across all test sets. Even with only 10 inference steps, which is significantly more efficient, it achieves a WER of 1.69 on *Regular zh*, compared to 2.28 from MaskGCT. On more challenging test sets, such as *Cross-lingual en2zh*, it reaches 4.66 (vs. 13.78 from F5-TTS), and on *Code-switching en*, it achieves 14.94 (vs. 35.35 from F5-TTS). In terms of SIM, both TaDiCodec-AR and TaDiCodec-MGM show clear advantages over recent systems such as FireRedTTS, SparkTTS, and Llasa. Their SIM scores are slightly lower than those of MaskGCT and CosyVoice 2, which operate at higher frame rates of 50 Hz and 25 Hz, respectively.

Table 6: **Results and RTF analysis for TTS model size scaling.**

| System | Model Size | RTF | Regular | | | | Articulatory | | | | Code-switching | | | | Cross-lingual | | | |
| | | | en | | zh | | en | | zh | | en | | zh | | zh2en | | zh2en | |
| | | | WER | SIM | WER | SIM | WER | SIM | WER | SIM | WER | SIM | WER | SIM | WER | SIM | WER | SIM |
| *Baseline systems* | | | | | | | | | | | | | | | | | | |
| CosyVoice 2 [41] | 0.5B | 0.47 | 2.89 | 0.66 | 1.29 | 0.76 | 8.63 | 0.66 | 7.60 | 0.74 | 28.32 | 0.59 | 38.39 | 0.75 | 9.98 | 0.67 | 7.59 | 0.53 |
| SparkTTS [6] | 0.5B | 0.59 | 2.50 | 0.57 | 1.78 | 0.66 | 10.19 | 0.57 | 13.37 | 0.65 | 15.12 | 0.46 | 16.86 | 0.65 | 9.73 | 0.58 | 4.88 | 0.40 |
| Llasa [5] | 1.0B | 0.42 | 3.94 | 0.58 | 8.02 | 0.64 | 11.36 | 0.55 | 21.20 | 0.58 | 17.56 | 0.46 | 26.98 | 0.59 | 26.47 | 0.49 | 9.18 | 0.41 |
| *Ours* | | | | | | | | | | | | | | | | | | |
| **TaDiCodec-MGM** | 0.6B | 0.12 | 3.69 | 0.65 | 1.51 | 0.75 | 10.67 | 0.63 | 8.97 | 0.71 | 14.76 | 0.57 | 20.01 | 0.73 | 9.95 | 0.65 | 4.75 | 0.48 |
| **TaDiCodec-AR-0.2B** | 0.2B | 0.20 | 7.68 | 0.64 | 1.48 | 0.74 | 16.06 | 0.63 | 12.54 | 0.70 | 16.38 | 0.56 | 23.91 | 0.72 | 13.40 | 0.64 | 4.26 | 0.48 |
| **TaDiCodec-AR-0.5B** | 0.5B | 0.22 | 3.88 | 0.65 | 1.15 | 0.75 | 12.09 | 0.63 | 9.04 | 0.70 | 13.58 | 0.57 | 17.10 | 0.73 | 8.79 | 0.64 | 4.07 | 0.48 |
| **TaDiCodec-AR-3B** | 3.0B | 0.25 | 3.24 | 0.65 | 1.23 | 0.75 | 8.34 | 0.63 | 8.52 | 0.70 | 11.31 | 0.57 | 15.47 | 0.73 | 7.85 | 0.65 | 3.99 | 0.48 |
| **TaDiCodec-AR-4B** **TaDiCodec-AR-4B w. vllm** | 4.0B | 0.29 0.13 | 2.28 | 0.65 | 1.19 | 0.75 | 8.23 | 0.63 | 8.74 | 0.70 | 9.16 | 0.57 | 16.09 | 0.73 | 7.67 | 0.64 | 2.91 | 0.48 |

Figure 3: **Performance gap between reconstruction and generation.** Each system includes both English and Chinese variants. Bars represent WER and SIM for reconstruction and generation.

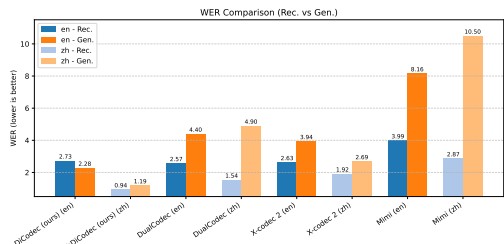

(a) WER gap between reconstruction and generation.

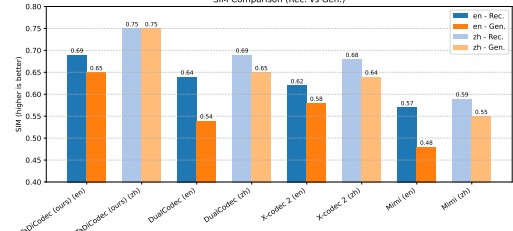

(b) SIM gap between reconstruction and generation.

**Model Size Scaling, Training and Inference Efficiency** We demonstrate that our 6.25 Hz tokenization is not only effective but also significantly **more efficient for both training and generation**. We further explore how scaling the model size affects both performance and efficiency. Results are shown in table 6. As described in the implementation details, we train all our TTS models for 300K steps. We find that the models achieve the optimal evaluation results at around 200K steps. All models can be trained in approximately one day under our setup, which uses 8 NVIDIA A100 GPUs with flash attention and bf16 precision. For inference efficiency, we measure using Real-Time Factor (RTF). We use a 5-second speech as a prompt to generate approximately 10 seconds of speech, sampling 5 times and taking the average. The experiments show that even with 4.0B parameters, our AR model can achieve an RTF of 0.29 without any deployment acceleration. With vLLM [76], the 4.0B AR model can achieve an RTF of 0.13. Additionally, the 0.6B TaDiCodec-MGM model achieves an RTF of 0.12. We also observe a reasonable improvement in performance with increasing model parameters, especially on challenging test sets (*Articulatory*, *Code-switching*, and *Cross-lingual*). Notably, our 0.5B model already matches or surpasses many state-of-the-art systems with an RTF of 0.22.

**Reconstruction and Generation Gap** In Figure 3, we present the performance gap between reconstruction and generation across multiple systems. Our proposed system, TaDiCodec, demonstrates a notably small performance gap: -16.5% for English WER (generation better than reconstruction), -5.8% for English SIM, +26.5% for Chinese WER, and -0.0% for Chinese SIM. These results indicate that TaDiCodec is highly generation-friendly—preserving most of the reconstruction quality during generation. In contrast, existing systems such as Mimi exhibit a much larger degradation (*e.g.*, -104.5% en WER gap and -265.9% zh WER gap), suggesting that they are less effective in transferring reconstruction capabilities to generation. This highlights the advantage of our design in ensuring consistency between reconstructed and generated outputs.

## 5 Conclusion

In this work, we introduce TaDiCodec, a novel speech tokenizer that injects textual information into the decoder and incorporates a prompt mechanism within an end-to-end diffusion autoencoder training framework. TaDiCodec achieves an extremely low frame rate of 6.25 Hz and a corresponding bitrate of 0.0875 kbps, using a single-layer codebook for 24 kHz speech. Beyond reconstruction, we apply TaDiCodec to zero-shot TTS using both AR and MGM, demonstrating its effectiveness,

efficiency, and suitability for generation. These results highlight TaDiCodec as a viable and innovative solution for speech language modeling.

# 6 Acknowledgment

The authors gratefully acknowledge the funding support from the National Natural Science Foundation of China (Grant No. 62376237), the Shenzhen Science and Technology Program (Grant No. ZDSYS20230626091302006), the Shenzhen Research Institute of Big Data (Internal Project Fund, Grant No. T00120230002), and the 2023 Shenzhen Stability Science Program. We would also like to thank the anonymous reviewers and the Area Chair for their insightful comments and valuable suggestions, which helped improve our paper.

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

# A Implementation Details

## A.1 Model Architecture

All our models follow the standard Transformer [51, 61] architecture, employ RoPE positional encoding [62] and the SiLU [77] activation function. The encoder and decoder of the tokenizer and MGM models use bidirectional attention, while the AR models adopt causal attention.

The TaDiCodec-AR-0.5B and TaDiCodec-AR-3B models are initialized from the textual LLMs `Qwen2.5-0.5B-Instruct` and `Qwen2.5-3B-Instruct` [65], respectively, while TaDiCodec-AR-4B is initialized from `Phi-3.5-mini-instruct` [64].

Table 7: **Model configurations.**

| Model | Hidden size | Intermediate size | Num. hidden layers | Num. attention heads | Num. key value heads | Num. parameters |
|---|---|---|---|---|---|---|
| TaDiCodec Encoder | 8 | 1024 | 4096 | 16 | 16 | ∼ 0.16B |
| TaDiCodec Decoder | 16 | 1024 | 4096 | 16 | 16 | ∼ 0.32B |
| TaDiCodec-AR-0.2B | 672 | 2048 | 24 | 14 | 2 | ∼ 0.25B |
| TaDiCodec-AR-0.5B | 896 | 4864 | 24 | 14 | 2 | ∼ 0.5B |
| TaDiCodec-AR-3B | 2048 | 11008 | 36 | 16 | 2 | ∼ 3B |
| TaDiCodec-AR-4B | 3072 | 8192 | 32 | 32 | 32 | ∼ 4B |
| TaDiCodec-MGM | 1280 | 5120 | 16 | 1 6 | 16 | ∼ 0.6B |

# B Flow Matching

We provide additional details of the flow matching framework used to train the diffusion decoder in TaDiCodec. Flow matching [35] defines a continuous transformation from a prior distribution (*e.g.*, Gaussian noise) to a target data distribution (*e.g.*, mel-spectrograms) by learning a time-dependent velocity field along an interpolated trajectory $x_t$.

While multiple interpolation strategies can be used to construct $x_t$, we adopt the *optimal transport path* formulation [35, 52], instantiated in this work as simple linear interpolation. Specifically, given a clean mel-spectrogram $x \in \mathbb{R}^{T \times d}$ and a noise sample $\epsilon \sim \mathcal{N}(0, I)$, we construct an intermediate sample as:

$$x_t = tx + (1 - t)\epsilon, \quad t \sim \text{Uniform}(0, 1), \tag{4}$$

where $t$ is sampled uniformly from $[0, 1]$, and $x_t$ denotes the noisy input at time $t$. The corresponding ground-truth velocity is the temporal derivative of $x_t$:

$$v = \frac{dx_t}{dt} = x - \epsilon. \tag{5}$$

The diffusion decoder $\mathcal{D}_\phi$ is trained to predict $v$, conditioned on the token sequence $q = Q(\mathcal{E}_\theta(x))$ and the associated text $x_{text}$, using the following objective:

$$\mathcal{L}_{\text{diff}} = \mathbb{E}_{(x, x_{text}), \epsilon, t} \left[ \left\| (x - \epsilon) - \mathcal{D}_\phi(q, x_t, t, x_{text}) \right\| \right]. \tag{6}$$

**Inference** At inference time, we start with a noise sample $x_0 = \epsilon \sim \mathcal{N}(0, I)$ and solve the ordinary differential equation:

$$\frac{dx_t}{dt} = \mathcal{D}_\phi(q, x_t, t, x_{text}) \tag{7}$$

from $t = 0$ to $t = 1$ using a simple Euler ODE solver over a discretized set of $N$ time steps.

Flow matching provides a stable and interpretable training signal by directly supervising the instantaneous direction in which a noisy sample $x_t$ should evolve to match the clean target $x$. In our setting, it enables effective training of the speech tokenizer under low bitrate constraints.

# C Binary Spherical Quantization

Binary Spherical Quantization (BSQ) [53] optimizes over an implicit codebook $C_{\text{BSQ}} = \left\{ -\frac{1}{\sqrt{L}}, \frac{1}{\sqrt{L}} \right\}^L$, which corresponds to the $L$-dimensional hypercube projected onto the unit sphere. Each corner $c_k \in C_{\text{BSQ}}$ represents a unique discrete token $k \in \{0, \dots, 2^L - 1\}$.

Given an encoder output $\mathcal{E}(\boldsymbol{x})$, we first obtain a low-dimensional latent sequence $\boldsymbol{h} \in \mathbb{R}^{T_q \times L}$ after linear projection. BSQ then projects each vector $\boldsymbol{h}_t$ in $\boldsymbol{h}$ onto the unit sphere:

$$\boldsymbol{u}_t = \frac{\boldsymbol{h}_t}{\|\boldsymbol{h}_t\|}, \tag{8}$$

and performs binary quantization independently on each dimension:

$$\hat{\boldsymbol{u}}_t = \frac{1}{\sqrt{L}} \operatorname{sign}(\boldsymbol{u}_t), \tag{9}$$

where $\operatorname{sign}(x)$ is the element-wise sign function, with $\operatorname{sign}(0)$ defined as 1 to ensure codewords lie on the unit sphere. To enable gradient-based training, BSQ uses the Straight-Through Estimator (STE) for backpropagation:

$$\operatorname{sign}_{\text{STE}}(x) = \operatorname{sg}(\operatorname{sign}(x) - x) + x, \tag{10}$$

where $\operatorname{sg}(\cdot)$ denotes the stop-gradient operation.

For each vector $\boldsymbol{h}_t$, the corresponding discrete token index is computed as:

$$k_t = \sum_{i=1}^{L} 1_{[\boldsymbol{h}_{t,i}>0]} \cdot 2^{i-1}, \tag{11}$$

where $1_{[\cdot]}$ is the indicator function. This efficient implicit code assignment scheme allows fast token computation and decoding via bitwise operations.

BSQ offers several appealing properties: it avoids the need for an explicit learnable codebook; its quantization error is bounded, allowing the entire system to be trained without a commitment loss [54].

In this work, we use $L = 14$, resulting in a codebook size of $2^{14} = 16384$.

# D  Masked Generative Models

In this section, we provide a brief introduction to masked generative models (MGMs) [7, 59, 78]. Let $\boldsymbol{x} = [y_1, y_2, \ldots, y_n]$ denote a discrete sequence of length $n$. At each time step $t$, we define the masked input as $\boldsymbol{x}_t = \boldsymbol{x} \odot \boldsymbol{m}_t$, where $\boldsymbol{m}_t = [m_{t,1}, m_{t,2}, \ldots, m_{t,n}]$ is a binary mask. Specifically, $x_i$ is replaced with a special [MASK] token if $m_{t,i} = 1$, and remains unchanged if $m_{t,i} = 0$. Each mask element $m_{t,i}$ is independently sampled from a Bernoulli distribution with parameter $\gamma(t)$, where $\gamma(t) \in (0, 1]$ is a masking schedule function (e.g., $\gamma(t) = \sin\left(\frac{\pi t}{2T}\right)$ for $t \in (0, T]$). The fully unmasked input is denoted by $\boldsymbol{x}_0 = \boldsymbol{x}$.

MGMs are trained to reconstruct the original sequence from partially observed inputs, conditioned on an optional context $\boldsymbol{c}$ (*e.g.*, in this paper, text $x_{text}$ is condition), by modeling the conditional distribution $p_\theta(\boldsymbol{x}_0 \mid \boldsymbol{x}_t, \boldsymbol{c})$. The model parameters $\theta$ are optimized by minimizing the expected marginal cross-entropy over the masked tokens:

$$\mathcal{L}_{\text{mask}} = -\mathbb{E}_{\boldsymbol{x},t,\boldsymbol{m}_t} \sum_{i=1}^{n} m_{t,i} \cdot \log p_\theta(y_i \mid \boldsymbol{x}_t, \boldsymbol{c}). \tag{12}$$

At inference time, MGMs generate tokens in parallel via iterative decoding. The process begins with a fully masked sequence $\boldsymbol{x}_T$. Assuming a total of $S$ decoding steps, at each step $j \in \{1, \ldots, S\}$, a prediction $\hat{\boldsymbol{x}}_0$ is sampled from $p_\theta(\boldsymbol{x}_0 \mid \boldsymbol{x}_{T-(j-1)\cdot\frac{T}{S}}, \boldsymbol{c})$. Then, $\lfloor n \cdot \gamma(T - j \cdot \frac{T}{S}) \rfloor$ tokens are selected based on confidence scores to be remasked, resulting in a new masked sequence $\boldsymbol{x}_{T-j\cdot\frac{T}{S}}$.

The confidence score for $\hat{y}_i$ in $\hat{\boldsymbol{x}}_0$ is given by $p_\theta(y_i \mid \boldsymbol{x}_{T-(j-1)\cdot\frac{T}{S}}, \boldsymbol{c})$ if the position $i$ was masked; otherwise, its score is set to 1, indicating that unmasked tokens will not be remasked. The $\lfloor n \cdot \gamma(T - j \cdot \frac{T}{S}) \rfloor$ tokens with the lowest confidence scores are selected for masking.

Note that the method for computing confidence scores is not unique. For example, [79] propose *Token-Critic*, a separate critic model trained to estimate token-wise confidence, thereby guiding the sampling process. In addition, [79, 80] suggest that masked generative modeling can be interpreted as a simplified form of discrete diffusion.

In this work, we develop MGM models for *text-to-token*. Given the low token rate of 6.25 Hz, the task is relatively easy to model, and 10 to 25 inference steps are sufficient to achieve good results.

# E Baselines

## E.1 Speech Tokenizer

**EnCodec** [19]  A Residual Vector Quantization (RVQ)-based neural audio codec operating at a frame rate of 75 Hz. We use two codebooks for inference, achieving a bitrate of 1.5 kbps. We use the official checkpoint[1].

**DAC** [21]  An improved VQGAN-based [79] codec that projects latent features onto a low-dimensional space (*e.g.*, 8 dimensions) prior to quantization. We reproduce two variants: one utilizing three codebooks at a 25 Hz frame rate, and the other a single codebook at a 75 Hz frame rate. Both configurations operate at a token rate of 75 Hz and achieve a bitrate of 0.75 kbps.

**SpeechTokenizer** [27]  Enhances first-layer speech tokens via semantic distillation using features from HuBERT [30]. This tokenizer operates at 50 Hz and we use two codebooks for inference. We use the official checkpoint[2].

**Mimi** [10]  Follows the design of SpeechTokenizer but utilizes WavLM [28] for semantic distillation. The tokenizer employs eight codebooks, each of size 2,048, at a 12.5 Hz frame rate, resulting in a bitrate of 1.1 kbps. We use the official checkpoint[3].

**DualCodec** [26]  A state-of-the-art, low-frame-rate, semantically-enhanced neural audio codec designed for speech generation. DualCodec directly encodes SSL representations [42] into first-layer codec tokens. It adopt a configuration with a 12.5 Hz token rate and a 8-layer codebook hierarchy. The first codebook contains 16,384 entries, while the remaining five each contain 4,096 entries, yielding a bitrate of 1.225 kbps. We use the official checkpoint[4].

**BiCodec** [6]  A semantically-enhanced tokenizer with a single-layer codebook. It discretizes audio into semantic tokens based on features from wav2vec 2.0 [29]. It operates at a token rate of 50 Hz with a codebook size of 8,192, achieving a bitrate of 0.65 kbps. We use the official checkpoint[5].

**X-codec 2** [5]  Employs a dual-encoder design: a semantic encoder based on Wav2Vec2-BERT [81] and an acoustic encoder for low-level acoustic features. Their outputs are concatenated prior to quantization. It operates at a token rate of 50 Hz with a codebook size of 65,536, yielding a bitrate of 0.8 kbps. We use the official checkpoint[6].

**WavTokenizer** [23]  A single-codebook tokenizer trained on 800K hours of mixed-domain audio. It operates at a 75 Hz token rate with a codebook size of 4,096, resulting in a bitrate of 0.9 kbps. We use the official checkpoint[7].

**BigCodec** [22]  A single-codebook tokenizer with scaled model size. It integrates sequential modules into convolutional architectures and applies low-dimensional quantization to enhance code utilization. It operates at an 80 Hz token rate with a codebook size of 8,192, yielding a bitrate of 1.04 kbps. We use the official checkpoint[8].

**TAAE** [24]  A transformer-based tokenizer using Finite Scalar Quantization (FSQ) [82] for speech tokenization. It operates at a 25 Hz token rate with a codebook size of 46,656, resulting in a bitrate of 0.4 kbps. We use the official implementation[9].

**SemantiCodec** [40]  Combines a semantic encoder (AudioMAE [83] with k-means clustering) and an acoustic encoder, featuring a diffusion decoder for reconstruction. It operates at a 50 Hz token rate, with codebook sizes of 16,384 (semantic) and 2,048 (acoustic), achieving a bitrate of 0.675 kbps. We use the official implementation[10].

---

[1]https://huggingface.co/facebook/encodec_24khz
[2]https://github.com/ZhangXInFD/SpeechTokenizer
[3]https://huggingface.co/kyutai/mimi
[4]https://pypi.org/project/dualcodec/0.1.2/
[5]https://github.com/SparkAudio/Spark-TTS
[6]https://huggingface.co/HKUSTAudio/xcodec2
[7]https://huggingface.co/novateur/WavTokenizer-large-speech-75token
[8]https://huggingface.co/Alethia/BigCodec
[9]https://github.com/Stability-AI/stable-codec
[10]https://github.com/haoheliu/SemantiCodec-inference

**Vevo Tokenizer [32]**  A two-stage tokenizer utilizing features from HuBERT [30], followed by VQ and a diffusion decoder. It employs a single codebook of size 8,192 at a 50 Hz token rate, resulting in a bitrate of 0.65 kbps. We use the official checkpoint[11].

**FireRedTTS Tokenizer [4]**  A single-codebook tokenizer trained in two stages. Transforms speech into semantic embeddings via features from HuBERT [30], followed by a ResNet-based encoder and quantization. It uses a 40 ms frame shift and a codebook size of 16,384. A global embedding is also incorporated, and decoding is performed using flow matching. Its implementation is available[12].

**CosyVoice Tokenizer [3]**  A single-codebook tokenizer trained in two stages. The encoder is initialized from an ASR model [36] and subsequently trained with a supervised loss. A flow matching model is used to predict mel-spectrograms. It operates at a 25 Hz token rate and 0.3 kbps bitrate. Its code is available[13].

**CosyVoice 2 Tokenizer [41]**  An improved version of CosyVoice that replaces VQ with FSQ. It operates at a 25 Hz token rate and 0.325 kbps bitrate. Its official implementation is available[14].

**Ints Tokenizer [68]**  Combines the DualCodec [26] semantic encoder with a flow matching decoder, similar to the CosyVoice variants. It uses a single codebook with 16,384 entries at a 12.5 Hz token rate, resulting in a bitrate of 0.175 kbps. The resulting TTS model, Ints, demonstrates state-of-the-art intelligibility [68].

## E.2 Zero-shot TTS

**F5-TTS [56]**  An open-source flow matching-based TTS systems. It follows E2 TTS [57] and uses a flow matching transformer [35, 55] to convert the text to acoustic features directly [56].

**MaskGCT [7]**  An open-source large-scale MGM-based TTS system that eliminates the need for explicit alignment information between text and speech supervision, as well as phone-level duration prediction. We use the official code and checkpoint[15] which is trained on Emilia [60].

**ARS [7]**  Introduced as an AR baseline by [7]. and referred to as "AR + SoundStorm" in the original paper [7]. It adopts a cascaded architecture, including the AR *text-to-token* and the NAR MGM *codec-to-waveform* [46].

**CosyVoice 2 [41]**  An open-source, large-scale zero-shot TTS system built upon an AR model initialized from `Qwen2.5-0.5B-Instruct`, which predicts speech codes extracted by the CosyVoice 2 tokenizer.

**FireRedTTS [4]**  An open-source, large-scale AR-based zero-shot TTS system, which predicts speech codes extracted by the FireRedTTS tokenizer.

**Ints [68]**  An open-source, large-scale zero-shot TTS system built upon an AR model initialized from `Phi-3.5-mini-instruct`, which predicts 12.5 Hz speech codes extracted by the Ints tokenizer.

**SparkTTS [6]**  An open-source, large-scale zero-shot TTS system built upon an AR model initialized from `Qwen2.5-0.5B-Instruct`, which predicts speech codes extracted by the BiCodec [6].

**Llasa [5]**  An open-source, large-scale zero-shot TTS system built upon an AR model initialized from `Llama3.2-1B` [84], which predicts speech codes extracted by the X-codec 2 [5].

# F  Evaluation

## F.1  Test Sets

***SeedTTS test-en***  We adopt a test set introduced in Seed-TTS [2], consisting of 1,000 samples drawn from English public corpora, including the Common Voice dataset [85]. We refer to this set as

---

[11] https://github.com/open-mmlab/Amphion/tree/main/models/vc/vevo
[12] https://github.com/FireRedTeam/FireRedTTS
[13] https://github.com/FunAudioLLM/CosyVoice
[14] https://github.com/FunAudioLLM/CosyVoice
[15] https://github.com/open-mmlab/Amphion/blob/main/models/tts/maskgct

"*Regular en*" and use it for zero-shot TTS evaluation (Table 5 and Table 6). Additionally, it is used for evaluating the performance of our tokenizer.

***SeedTTS test-zh***    We adopt a test set introduced in Seed-TTS, comprising 2,000 samples drawn from Chinese public corpora, including the DiDiSpeech dataset [86]. We denote it as "*Regular zh*" for zero-shot TTS evaluation.

***Articulatory en, Articulatory zh***    These sets are introduced in [68] and contain tongue twisters and repeated texts. For Chinese, the *SeedTTS test-hard* set is used directly. For English, reference speech prompts are taken from *SeedTTS test-en*, while the corresponding articulatory texts are constructed using Deepseek-V3 [87] to match the style of *SeedTTS test-hard*. Each set contains 400 samples. An example:

> *Prompt text:*
> Salmon is one of the most popular fish and very delicious, though usually not sustainable.
>
> *Target text:*
> A big black bug bit a big black bear, but the big black bear bled black blood from the bite.

***Code-switching en, Code-switching zh***    These sets are introduced in [68], consist of target texts that mix English and Chinese. Based on *SeedTTS test-en* and *test-zh*, the reference speech prompts are kept unchanged, while Deepseek-V3 is employed to convert the texts into a code-switching format. Each set contains 500 samples. An example:

> *Prompt text:*
> 创下奥运史上拒绝奥运圣火入境的首例。
>
> *Target text:*
> 在他 execution 之后 Ogilvie 的 followers 被 rounded up 并 put in jail.

***Cross-lingual zh2en, Cross-lingual en2zh***    These sets are introduced in [68], two types of cross-lingual samples are constructed: *zh2en* and *en2zh*, each comprising 500 samples. The *zh2en* set pairs Chinese reference speech from *SeedTTS test-zh* with English target text from *SeedTTS test-en*, while the *en2zh* set follows the reverse configuration. Each set contains 500 samples. An example:

> *Prompt text:*
> 调整海外购买住宅征收额外印花税率，从百分之三调整到百分之七而言。
>
> *Target text:*
> The recluse from Lithuania and his compatriot were making up stories about mermaids and fays.

***Multilingual test sets***    We additionally construct four multilingual test sets to evaluate tokenizer reconstruction in non-English languages, including French (fr), German (de), Japanese (ja), and Korean (ko). For each language, we randomly sample 300 utterances from Common Voice [85].

### F.2    Objective Evaluation

**Frame Rate, Token Rate, Bitrate**    Frame rate means the speech is compressed into how many frames per second (measured in Hz), while each frame may contain multiple tokens; token rate refers to how many discrete tokens are produced per second; bitrate indicates the total amount of information retained, computed as token rate multiplied by the number of bits per token (measured in kbps), and reflects the overall compression level of the tokenizer.

For example, suppose a speech tokenizer operates at a frame rate of 25 Hz, meaning the input audio is compressed into 25 frames per second. If each frame contains 2 codebook tokens (*i.e.*, 2 layers of quantization), and each codebook has a size of 2048 (requiring 11 bits per token since $2^{11} = 2048$), then:

- **Token Rate** = 25 frames/sec × 2 tokens/frame = 50 tokens/sec
- **Bitrate** = 50 tokens/sec × 11 bits/token = 550 bps = 0.55 kbps

This means the speech is represented with a bitrate of 0.55 kbps, indicating a high compression level while retaining discrete structure for downstream modeling.

**WER**    Word Error Rate (WER) is employed to assess the intelligibility of reconstructed or generated speech. We adopt two automatic speech recognition (ASR) models for WER computation: `whisper-large-v3`[16] [36] and `paraformer-zh`[17] [37]. The former is used for non-Chinese utterances, while the latter is applied to Chinese speech, following established practices in recent studies [2, 3, 7, 58].

**SIM**    Speaker similarity (SIM) is computed as the cosine similarity between speaker embeddings extracted from the prompt and the generated utterance. We use the `WavLM-TDNN` model[18] [28] for speaker embedding extraction, following [1, 2, 7, 45, 55].

**UTMOS**    Speech naturalness and perceptual quality are evaluated using UTMOS [88], a Mean Opinion Score (MOS) prediction system. UTMOS combines ensemble learning of strong and weak learners: the strong learners are fine-tuned self-supervised learning (SSL) models with architectural enhancements, while the weak learners apply lightweight regression on SSL features. We use the official UTMOS checkpoint[19].

### F.3 Subject Evaluation

We conduct a subjective evaluation of speech tokenizers in terms of audio quality using the Comparative Mean Opinion Score (CMOS):

- **System Interface**: Users listen to two speech samples, A and B, to compare their speech quality.
- **Instruction**: Participants are asked, "Which speech has better audio quality?".
- **Evaluation Criteria**: Five response options: A +2 (Sample A has much better audio quality), A +1 (Sample A has slightly better audio quality), Tie (Both have equal audio quality), B +1 (Sample B has slightly better audio quality), and B +2 (Sample B has much better audio quality).

Figure 4 shows a shotscreen of the evaluation system.

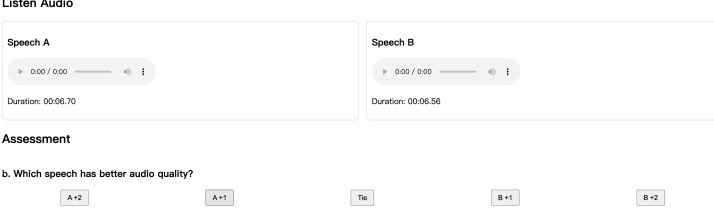

Figure 4: **Shotscreen of the subjective evaluation system.**

We randomly select 40 samples from an in-the-wild dataset. Each of the six systems: Ours, X-codec 2, DualCodec, WavTokenizer, Mimi, and Ground Truth, generates all 40 samples. For evaluation, each baseline system is compared against ours, resulting in a total of $40 \times 5 = 200$ sample pairs. Each pair is evaluated by two human listeners.

## G    Limitations and Future Work

TaDiCodec achieves an extremely low frame rate of 6.25 Hz and a corresponding bitrate of 0.0875 kbps using a single-layer codebook for 24 kHz speech compression, while demonstrating strong

---

[16]https://huggingface.co/openai/whisper-large-v3

[17]https://huggingface.co/funasr/paraformer-zh

[18]https://github.com/microsoft/UniSpeech/tree/main/downstreams/speaker_verification

[19]https://huggingface.co/spaces/sarulab-speech/UTMOS-demo

performance in both reconstruction and text-to-speech tasks in terms of intelligibility, speaker similarity, and speech quality. There remains room for improvement and several promising directions for future work: 1) TaDiCodec employs a diffusion autoencoder for tokenization and de-tokenization, which involves multiple steps during inference. Compared to GAN-based tokenizers, this results in higher decoding latency. Future work may explore distillation or more powerful generative models to enable single-step inference while maintaining performance. 2) While TaDiCodec has shown its effectiveness for speech language modeling through zero-shot TTS, it is worth further evaluating its applicability in speech understanding and dialogue systems. 3) TaDiCodec currently requires text input for the decoder. It would be valuable to explore unified models that can transcribe, tokenize, and reconstruct speech simultaneously, enabling one model for joint understanding, compression, and reconstruction.

## H    Broader Impacts

Our model enables high-quality speech modeling, which can benefit applications such as personalized speech interfaces, speech restoration, and accessibility tools. However, it also poses risks of misuse, including voice spoofing and unauthorized impersonation. These risks are particularly concerning in scenarios involving biometric authentication or deceptive media. To prevent misuse, we advocate for the development of reliable deepfake detection tools, watermarking methods for synthetic speech, and reporting mechanisms to flag suspected abuse.

