# OpenReview forum: "TaDiCodec: Text-aware Diffusion Speech Tokenizer for Speech Language Modeling"
_NeurIPS.cc/2025/Conference — NeurIPS 2025 poster_

### Official Review · Reviewer_WHX2 · 2025-07-02

**Clarity:** 2
**Significance:** 3
**Originality:** 3
**Rating:** 4
**Confidence:** 4

**Summary:**

The paper proposes a text-aware, diffusion-based codec model that uses a diffusion decoder to generate speech. It can be trained with a single stage and achieves a low bitrate compared to other methods. They also show that the model has a smaller reconstruction-generation gap using zero-shot TTS. From the zero-shot TTS result, auto-regressive models trained with the codec can achieve better performance than other prior work.

**Questions:**

1. Can the authors provide an ablation study on the text-aware component?
2. From the reconstruction demo on the demo page, it seems the codec has some subtle changes to the prosody. As the codec is trained as a generative model, it makes sense to involve some randomness. Is it possible to introduce a metric to quantify that?

**Ethical Concerns:**

["NO or VERY MINOR ethics concerns only"]

**Final Justification:**

I accepted the author's rebuttal and would like to retain my rating.

**Limitations:**

They are put in the appendix. I'm not sure if it's okay.

**Quality:**

2

**Strengths And Weaknesses:**

The paper proposes an interesting approach for speech codecs. By adding text conditions and prompts to the diffusion decoder, it can achieve a lower bitrate than prior work. It also shows promising zero-shot TTS results.

However, some claims from the paper seem wrong. Firstly, they refer the auto-regressive zero-shot TTS as "speech language models", which should include a broader range of models, such as Moshi-style text-speech LMs. So "improving the reconstruction-generation gap for SLMs" seems to be overclaiming. Secondly, while they show a promising result on the zero-shot TTS task, there is no evidence on whether it benefits from the text-aware and the prompt, as other approaches don't have that. It will be beneficial to have ablation study on the text-aware component. A minor point: the results part directly refer to the number in the table, which is hard to follow.

---

> ### Author Rebuttal · Authors · 2025-07-30
>
> Dear reviewer, we want to thank for your carefully reading and providing the constructive comments! Below we address the concerns mentioned in the review.
>
> `Weakness 1: However, some claims from the paper seem wrong. Firstly, they refer the auto-regressive zero-shot TTS as "speech language models", which should include a broader range of models, such as Moshi-style text-speech LMs. So "improving the reconstruction-generation gap for SLMs" seems to be overclaiming.`
>
> Thank you for your suggestion. We acknowledge that the term speech language modeling can have broader interpretations depending on context. In our work, we use speech language modeling to specifically refer to generation speech tokens with language models, particularly in the context of text-conditional speech generation. This definition has also been adopted in prior works on speech tokenizers [1, 2, 3, 4], which similarly evaluate their models primarily on language-model-based TTS tasks.
>
> We fully understand your concern about potential overclaiming, and we commit to clarifying our definition of speech language modeling in the revised version of the paper to avoid overclaiming.
>
> Additionally, we are actively exploring the use of TaDiCodec (as the output) for building Moshi- or GLM4-Voice-style dialog speech LLMs, where both text and speech tokens are generated jointly. These models can decode speech outputs via the TaDiCodec decoder, which also motivates our tokenizer design.
>
> `Weakness 2: while they show a promising result on the zero-shot TTS task, there is no evidence on whether it benefits from the text-aware and the prompt, as other approaches don't have that. It will be beneficial to have ablation study on the text-aware component.`
>
> We compare all baseline TTS models in a zero-shot setting, where a prompt is provided for each sample to guide the generation. In fact, most of these baselines  require prompts for their language models and diffusion decoders (if such components are present). In our case, we additionally provide text to the decoder, as this is a core part of our design.
>
> For the ablation study on the text-aware component, we provide additional experiments demonstrating its impact in our response to Question 1.
>
> `Question 1: Can the authors provide an ablation study on the text-aware component?`
>
> We have conducted additional experiments to study the impact of the text-aware decoder. Since the core design of our tokenizer is to achieve high-quality reconstruction under extremely low token rates with the diffusion autoencoder and the guidance of text information, we started with the most basic setting: a GAN-based speech tokenizer. As shown below, the GAN-based model struggles to reconstruct intelligible speech at such low token rates and suffers from unstable training behavior compared to diffusion-based models.
>
> We then replaced the tokenizer with an end-to-end trained diffusion autoencoder without any textual conditioning. We observe that both speech quality and similarity improve significantly, and the WER also decreases noticeably, though it still remains relatively high at this stage.
>
> Finally, we introduced the text-aware decoder, which further improved performance across all metrics. Introducing the text-aware decoder significantly reduces the representational burden on the tokens, allowing them to focus on learning residual information beyond the text, while still potentially capturing some textual content, under very low token-rate settings. This design finally makes accurate reconstruction.
>
> | System                        | WER | SIM  | UTMOS |
> |------------------------------|---------|------|-------|
> | GAN                          | 65.7    | 0.38 | 2.97  |
> | Diffusion                    | 22.5    | 0.59 | 3.23  |
> | Diffusion + text-aware decoder | 2.73    | 0.69 | 3.73  |
>
> `Question 2: From the reconstruction demo on the demo page, it seems the codec has some subtle changes to the prosody. As the codec is trained as a generative model, it makes sense to involve some randomness. Is it possible to introduce a metric to quantify that?`
>
> We provide additional evaluation metrics to measure prosodic deviation between the generated speech and the ground truth. Specifically, we compute the F0 Correlation (FPC) over all samples in the test set to assess overall prosody preservation.
>
> To quantify the randomness introduced by the generative model, particularly due to the initial gaussian noise in flow matching models, we generate each sample five times and compute the standard deviation of FPC across these generations. We use the average standard deviation across the dataset as an indicator of prosodic variability introduced by sampling stochasticity.
>
> | Model               | Bitrate | Mean FPC | Mean FPC Std (generating 5 times) |
> |---------------------|---------|----------|------------------------------------|
> | DualCodec           | 1.225   | 0.77     | 0.00                              |
> | CosyVoice Tokenizer | 0.65    | 0.67     | 0.028                              |
> | Vevo Tokenizer      | 0.175   | 0.68     | 0.031                              |
> | TaDiCodec           | 0.0875  | 0.71     | 0.039                              |
>
>
> We can observe that DualCodec achieves the highest mean FPC and zero mean FPC std, which can be attributed to its discriminative modeling approach and highest bitrate, allowing it to preserve prosody more precisely. In contrast, CosyVoice Tokenizer, Vevo Tokenizer, and TaDiCodec yield similar lower mean FPC scores,while TaDiCodec exhibits a higher mean FPC Std, likely due to its lower bitrate.
>
> By the way, from a perceptual perspective, we observe that minor variations in prosody tend to occur more across different samples, rather than across multiple generations of the same sample. That is, the differences introduced by stochastic sampling (of the  initial gaussian noise) are generally subtle.
>
> Thanks again for your valuable comments, we would be grateful if we could hear your feedback regarding our answers. We would be happy to answer and discuss if you have further comments.
>
> Reference:
>
> > [1] Zhang X, Zhang D, Li S, et al. Speechtokenizer: Unified speech tokenizer for speech large language models[J]. arXiv preprint arXiv:2308.16692, 2023.
>
> > [2] Ji S, Jiang Z, Wang W, et al. Wavtokenizer: an efficient acoustic discrete codec tokenizer for audio language modeling[J]. arXiv preprint arXiv:2408.16532, 2024.
>
> > [3] Ye Z, Sun P, Lei J, et al. Codec does matter: Exploring the semantic shortcoming of codec for audio language model[C]//Proceedings of the AAAI Conference on Artificial Intelligence. 2025, 39(24): 25697-25705.
>
> > [4] Li J, Lin X, Li Z, et al. DualCodec: A Low-Frame-Rate, Semantically-Enhanced Neural Audio Codec for Speech Generation[J]. arXiv preprint arXiv:2505.13000, 2025.

---

### Official Review · Reviewer_rmsF · 2025-07-03

**Clarity:** 2
**Significance:** 2
**Originality:** 2
**Rating:** 4
**Confidence:** 5

**Summary:**

This work introduces TaDiCodec, a speech tokenizer that achieves low frame rate and bitrate using a single-layer codebook. The key idea is the integration of text information into a diffusion-based decoder. Unlike existing approaches that require multi-stage training, external pre-trained models for semantic distillation, TaDiCodec uses a unified diffusion autoencoder that jointly performs quantization and reconstruction. The authors showed the tokenizer's effectiveness through speech reconstruction tasks and zero-shot text-to-speech.

**Questions:**

- How does the model perform when text input is noisy or contains errors (e.g., from imperfect ASR)? Have you evaluated robustness to text quality degradation?
- Can you provide more insight into why text conditioning enables such extreme compression? Is there a theoretical formulation that explains the information-theoretic relationship between text and the required speech token rate?
- Have you explored distillation methods (e.g., consistency models) to reduce the number of diffusion steps to 1-2 while maintaining quality? This could significantly improve inference efficiency.

**Ethical Concerns:**

["NO or VERY MINOR ethics concerns only"]

**Final Justification:**

Some of my concerns were addressed, including the additional DASB results, robustness to ASR transcript quality, and clarification on codebook utilization. I appreciate the clarification regarding missing baselines (e.g., DM-Codec) and the provided rationale for excluding Single-Codec.

That said, several issues remain insufficiently addressed:

* **Benchmarking**: While DASB was partially evaluated, no results were shown for Codec-SUPERB. ARCH was a typo. Thank you for the clarification. I was referring to comprehensive evaluation suites such as HEAR or similar holistic benchmarks that assess general-purpose capabilities.

* **Model comparison fairness**: Training data discrepancies across baselines (100k+ vs. others) remain a concern. Simply stating that other models "likely" use similar data pipelines is speculative and does not eliminate the concern regarding fair comparison.

* **Evaluation scope**: The evaluation still focuses only on speech reconstruction and generation. It omits important speech understanding tasks, which are essential to validate tokenizers for broader use beyond TTS.

* **Failure case analysis**: The discussion on long utterances and singing is useful, but mostly anecdotal. A more systematic quantitative analysis would strengthen the claims.

* **Ablations**: Still missing key ablations, such as on text-speech misalignment and architectural alternatives. While the authors argue for a minimal design, this design choice itself warrants ablation to verify its effectiveness and generalizability.

* **Clarification on DM-Codec**: I respectfully disagree with the authors’ claim that DM-Codec's performance is “comparable to SpeechTokenizer.” DM-Codec demonstrates strong performance across both generation and understanding tasks, even at lower token rates, depending on the configuration. Its RVQ depth does not necessarily imply a higher effective bitrate, as token pruning and entropy-based compression techniques are often used. The dismissal without deeper analysis seems premature.

**Conclusion**: This is a promising direction and a well-written paper with a clean design and competitive results on TTS tasks. However, the limited evaluation scope, lack of comprehensive benchmarking, and insufficient ablation weaken the overall strength of the contributions. I would increase my rating to **borderline accept**. While I recognize some contributions, I do not see enough depth to champion this paper during AC discussions, specifically following the NeurIPS acceptance bar.

Thanks again.

**Limitations:**

The authors adequately address limitations in Appendix G

**Paper Formatting Concerns:**

- Seems like author extensively used -ve : Figure-2, Eq-2 etc
- Most of the table font is tiny. Barely readable. Table formatting can be improved significantly

**Quality:**

2

**Strengths And Weaknesses:**

**Strengths:**
- Incorporating text-aware decoding into the diffusion process, eliminating the need for multi-stage pipelines
- Achieving 6.25 Hz frame rate with 0.0875 kbps bitrate while maintaining competitive reconstruction quality
- WER of 2.28 on SeedTTS test-en and 1.19 on test-zh showed SOTA performance

**Weaknesses**
- Only compares with Encodec, DAC, FACodec, SpeechTokenizer, and SemanticCodec. Missing some recent works: WavTokenizer, StableCodec, Single-Codec, BiCodec, and DM-Codec
- The paper lacks evaluation on established comprehensive benchmarks: Codec-SUPERB, DASB, or ARCH benchmarks. Any particular reasons to ignore these benchmarks?
- Codebook utilization rate is not reported, which is crucial for understanding quantization efficiency
- No analysis of model size or memory requirements compared to baselines
- TaDiCodec uses 100k hours while some baselines use different amounts. Is it a fair comparison?
- Only 5,000 samples for test-en and test-zh, which may be insufficient for robust statistical significance analysis
- Insufficient Ablation Studies:
    - No ablation on the effect of text quality (e.g., using noisy ASR transcripts vs. ground truth)
    - No analysis of performance degradation when text-speech misalignment
    - Missing ablation on different text encoder architectures or pre-trained models
- Limited Analysis of Failure Cases:
    - Performance on edge cases (very long utterances, extreme speaking rates)
    - Comparison of failure modes with text-free tokenizers
- No evaluation on speech understanding tasks. Why evaluated only evaluated on TTS tasks?

Please feel free to point me if I missed anything. Thanks!

---

> ### Author Rebuttal · Authors · 2025-07-30
>
> Dear reviewer, we want to thank for your carefully reading and providing the constructive comments! Below we address the concerns mentioned in the review.
>
> `Weakness 1: Missing some recent works: WavTokenizer, StableCodec, Single-Codec, BiCodec, and DM-Codec.`
>
> In Table 1 of our paper, which presents the main results of our tokenizers, we compare against several representative baselines, including **WavTokenizer, BiCodec, StableCodec (TAAE)**, as well as  some other recent models such as **DualCodec, x-codec 2, Mimi, BigCodec, Vevo Tokenizer, FireRedTTS Tokenizer, CosyVoice 1/2 Tokenizer, and Ints Tokenizer**.
>
> For Single-Codec, we did not include results as the model weight and inference code are not publicly available. We did not reproduce the model as our benchmark already includes a sufficiently diverse set of strong baselines.
>
> Regarding DM-Codec, we were not previously aware of this work and will add the appropriate citation in our revision. While it adopts 8 layers of RVQ (which means it has a much higner token rate), its performance is comparable to that of SpeechTokenizer.
>
> `Weakness 2: The paper lacks evaluation on established comprehensive benchmarks: Codec-SUPERB, DASB, or ARCH benchmarks...`
>
> Our work primarily focuses on designing speech tokenizers tailored for speech generation with language models or as the output of speech language models. Under this setting, we explore how to compress speech representations as much as possible. So, our evaluation mainly targets speech reconstruction and speech generation with language modeling.
>
> To address the reviewer's suggestion, we additionally evaluate our codec on the DASB benchmark [1]. Due to time constraints, we select two representative tasks: emotion recognition and speaker verification. Following the official DASB protocol, we train models using our codec embeddings (after projection) and compare the performance against SpeechTokenizer as reported in the original DASB paper:
>
> | Model | ER (Acc) | SV (EER) |
> |-|-|-|
> | TaDiCodec | 54.5 | 16.7 |
> | SpeechTokenizer | 49.1 | 20.4 |
>
> Furthermore, we could not identify a benchmark named **ARCH** that is publicly available or specifically related to speech codec evaluation. Could you please provide more details about it? We would be happy to look into it!
>
> `Weakness 3: Codebook utilization rate is not reported.`
>
> Thank you for the suggestion. We adopt BSQ for quantization, which significantly alleviates the codebook collapse issue. In our experiments, with a codebook size of 16,384, our method achieves a utilization rate of over 99%.
>
> `Weakness 4: No analysis of model size or memory requirements compared to baselines.`
>
> We provide detailed model configurations in Appendix A.1, along with an ablation study on tokenizer model size in Table 4, and analysis of model size and real-time factor (RTF) for TTS in Table 6.
>
> Since the core design of our work leverages a text-aware module and a diffusion autoencoder to enable high-quality reconstruction at an extremely low token rate (6.25 Hz), while maintaining efficient generation within the full speech generation with language modeling pipeline, we believe that the results and analysis presented in Table 6 sufficiently support our claims regarding efficiency and model compactness.
>
> `Weakness 5: TaDiCodec uses 100k hours while some baselines use different amounts. Is it a fair comparison?`
>
> Indeed, we are not able to ensure that all compared tokenizers are trained on exactly the same dataset. Our model, TaDiCodec, is trained on the Emilia dataset consisting of 100k hours of speech. A significant portion of the baselines are also trained on Emilia or its supersets (which implies they use more training data including Emilia), or they are trained on datasets larger than 100k hours collected with similar data pipelines. These include: Mimi , DualCodec (Emilia), X-codec 2 (Emilia + MLS), Vevo Tokenizer (Emilia), FireRedTTS Tokenizer (200k+ hours), CosyVoice 1/2 Tokenizer (170k+ hours), and Ints Tokenizer (Emilia).
>
> `Weakness 6: Only 5,000 samples for test-en and test-zh, which may be insufficient for robust statistical significance analysis.`
>
> We follow standard benchmarks commonly used in prior work on speech generation, and **provide a detailed discussion of our evaluation datasets in Appendix F**.
>
> Specifically, for tokenizer evaluation, we use SeedTTS test-en (1,000 samples) and test-zh (2,000 samples), along with multilingual test sets in French, German, Japanese, and Korean, each consisting of 300 utterances randomly sampled from Common Voice.
>
> For TTS evaluation, in addition to SeedTTS test-en and test-zh, we include 400 samples from Articulatory-en, 400 from Articulatory-zh, 500 from Code-switching-en, 500 from Code-switching-zh, 500 from Cross-lingual-zh2en, and 500 from Cross-lingual-en2zh. While many previous works only evaluate on SeedTTS test-en/test-zh or LibriSpeech.  In fact, the total number of evaluation samples used in our work exceeds that of most related papers.
>
> `Weakness 7: Insufficient Ablation Studies.`
>
> Thank you for the valuable suggestion! We are more than happy to provide additional analysis as follows:
>
> **The effect of text quality and performance degradation when text-speech misalignment**:
>
> We use ground-truth text during inference. To assess the robustness of our system to imperfect text input, we additionally provide results using transcripts generated by Whisper-large-v3. On the SeedTTS test-en set, Whisper-large-v3 achieves a low WER of 2.15. We observe a slight degradation in WER when using ASR transcripts (from 2.73 to 3.24), while SIM and UTMOS remain nearly unchanged. This aligns with our motivation: modern ASR systems are already powerful enough for high-quality text transcription. Moreover, in the speech generation setting, the target text is typically predefined or user-specified, and thus always accessible. Therefore, using ground-truth text for evaluation does not affect the fairness or validity of our TTS results.
>
> | System | WER  | SIM  | UTMOS |
> |-|-|-|-|
> | TaDiCodec (with ground-truth text) | 2.73 | 0.69 | 3.73   |
> | TaDiCodec (with Whisper-large-v3 text)      | 3.24 | 0.69 | 3.72   |
>
> **Ablation on different text encoder architectures**:
>
> In fact, our work does not involve any dedicated text encoder for either the tokenizer or the TTS models. We follow a minimal design principle: for both the tokenizer and the TTS system, the text is simply treated as a prefix of the speech token sequence and fed jointly into a standard Transformer architecture using self-attention. No additional modules are used to separately process the text input (e.g., via a dedicated text encoder). While incorporating such components may offer potential improvements, this direction is orthogonal to our core design goals.
>
> `Weakness 8: Limited Analysis of Failure Cases.`
>
> In our demo page, we showcase several challenging cases such as singing and utterances with significant pitch variation, demonstrating that our method performs better than other baselines (with low token rates) in these scenarios.
> As for very long speech samples, we observe that performance degrades when the model processes utterances longer than 40 seconds in a single pass. The reason is the training data primarily consists of utterances shorter than 30 seconds. However, this issue can be mitigated by chunking long utterances into segments (e.g., 30 seconds each), which is straightforward to implement.
>
> `Weakness 9: No evaluation on speech understanding tasks. Why evaluated only evaluated on TTS tasks?`
>
> We provide a detailed explanation in our response to Weakness 2.
>
> `Question 1: How does the model perform when text input is noisy or contains errors (e.g., from imperfect ASR)? Have you evaluated robustness to text quality degradation?`
>
> We provide a detailed explanation in our response to Weakness 7.
>
> `Question 2: Can you provide more insight into why text conditioning enables such extreme compression? Is there a theoretical formulation that explains the information-theoretic relationship between text and the required speech token rate?`
>
> We're happy to provide more high-level insights. This is closely related to the **question raised in question 1 of reviewer rwbf**.
>
> TaDiCodec is inspired by recent NAR TTS systems such as E2TTS, which have demonstrated that it is possible to generate speech features directly from text using diffusion models. However, these models operate on high-dimensional continuous acoustic representations, which are often difficult to model accurately and comprehensively.
>
> In contrast, TaDiCodec introduces discrete tokens that encode essential information beyond semantics, including prosody, speaker characteristics, and text-speech alignment, which are critical for accurate reconstruction. Additionally, the powerful generative capacity of the diffusion decoder allows it to synthesize fine-grained acoustic details that do not need to be explicitly encoded in the tokens. This design enables TaDiCodec to achieve high-quality reconstruction and more accurately capture the target speech distribution.
>
> Due to space limitations, we provide a more detailed discussion in our response to question 1 of reviewer rwbf. Please refer to it for further insights.
>
> `Question 3: Have you explored distillation methods (e.g., consistency models) to reduce the number of diffusion steps to 1-2 while maintaining quality? This could significantly improve inference efficiency.`
>
> It's a very promising direction, and we are actively exploring this problem by finetuning the decoder through distillation. We plan to share results in future work.
>
> Thanks again for your valuable comments, we would be grateful if we could hear your feedback regarding our answers. We would be happy to answer and discuss if you have further comments.
>
> Reference
>
> > [1] Mousavi P, Della Libera L, Duret J, et al. Dasb-discrete audio and speech benchmark[J]. arXiv preprint arXiv:2406.14294, 2024.

---

> > ### Comment · Reviewer_rmsF · 2025-08-05
> > **Borderline Accept: Promising Direction but Lacks Comprehensive Evaluation**
> >
> > Thank you for the detailed rebuttal. Apologies for the late reply. I had a medical emergency.
> >
> > Some of my concerns were addressed, including the additional DASB results, robustness to ASR transcript quality, and clarification on codebook utilization. I appreciate the clarification regarding missing baselines (e.g., DM-Codec) and the provided rationale for excluding Single-Codec.
> >
> > That said, several issues remain insufficiently addressed:
> >
> > * **Benchmarking**: While DASB was partially evaluated, no results were shown for Codec-SUPERB. ARCH was a typo. Thank you for the clarification. I was referring to comprehensive evaluation suites such as HEAR or similar holistic benchmarks that assess general-purpose capabilities.
> >
> > * **Model comparison fairness**: Training data discrepancies across baselines (100k+ vs. others) remain a concern. Simply stating that other models "likely" use similar data pipelines is speculative and does not eliminate the concern regarding fair comparison.
> >
> > * **Evaluation scope**: The evaluation still focuses only on speech reconstruction and generation. It omits important speech understanding tasks, which are essential to validate tokenizers for broader use beyond TTS.
> >
> > * **Failure case analysis**: The discussion on long utterances and singing is useful, but mostly anecdotal. A more systematic quantitative analysis would strengthen the claims.
> >
> > * **Ablations**: Still missing key ablations, such as on text-speech misalignment and architectural alternatives. While the authors argue for a minimal design, this design choice itself warrants ablation to verify its effectiveness and generalizability.
> >
> > * **Clarification on DM-Codec**: I respectfully disagree with the authors’ claim that DM-Codec's performance is “comparable to SpeechTokenizer.” DM-Codec demonstrates strong performance across both generation and understanding tasks, even at lower token rates, depending on the configuration. Its RVQ depth does not necessarily imply a higher effective bitrate, as token pruning and entropy-based compression techniques are often used. The dismissal without deeper analysis seems premature.
> >
> > **Conclusion**: This is a promising direction and a well-written paper with a clean design and competitive results on TTS tasks. However, the limited evaluation scope, lack of comprehensive benchmarking, and insufficient ablation weaken the overall strength of the contributions. I would increase my rating to **borderline accept**. While I recognize some contributions, I do not see enough depth to champion this paper during AC discussions, specifically following the NeurIPS acceptance bar.
> >
> > Thanks again.

---

> ### Author Response · Authors · 2025-08-05
>
> Thank you very much for your response and your consideration of raising the score. We understand that you still have some remaining concerns, and we sincerely appreciate your thoughtful feedback. We will strive to address these issues in our future work. Meanwhile, we would also like to further clarify some of the questions.
>
> For **model comparison**, DualCodec, X-codec 2, Vevo Tokenizer, and Ints Tokenizer were all trained on Emilia. We will explicitly indicate the training datasets used for each baseline in the table of our paper to ensure a fairer and more transparent comparison.
>
> For "Failure case analysis", we provide some quantitative analysis on the slight prosodic variations introduced by randomness in the response to **Reviewer WHX2's Question 2**.
>
> For **Ablations**, we provide the ablation of the effect of text quality and performance degradation when text-speech misalignment in the response to **Weakness 7: Insufficient Ablation Studies**, we also provide ablation of the text-aware component in the response to **Reviewer WHX2's Question 1**. We provide it here:
>
> System | WER | SIM | UTMOS
> -|-|-|-
> without text-aware decoder |	22.5	| 0.59 | 3.23
> with text-aware decoder	| 2.73	| 0.69 | 3.73
>
> Thank you again for your valuable feedback!

---

### Official Review · Reviewer_rwbf · 2025-07-03

**Clarity:** 2
**Significance:** 2
**Originality:** 3
**Rating:** 4
**Confidence:** 5

**Summary:**

This paper introduces TaDiCodec, a novel speech tokenizer designed for speech language modeling that achieves an extremely low frame rate of 6.25 Hz and bitrate of 0.0875 kbps using a single-layer codebook. The key innovation is the use of an end-to-end diffusion autoencoder with text-aware decoding, eliminating the need for multi-stage training, external pre-trained models for semantic distillation, or complex adversarial objectives.

**Questions:**

1. Regarding the diffusion decoder, since text and prompt speech can already achieve zero-shot TTS through NAR diffusion models [1][2], I am curious about what information is actually being learned by the 6.25 tokens per second. Are these tokens encoding the residual information, maybe speech - text - global? Providing more analysis on this point would add valuable insights to the paper.

2.  In Figure 3a, the WER of reconstruction is generally better than that of generation, which is reasonable because reconstruction is based on the ground truth, while generation relies on the model’s learned distribution of real data. Therefore, the reconstruction of gt should theoretically serve as an upper bound for generation. That’s why I am very curious about, for TaDiCodec (ours) (en), the generation WER is even better than reconstruction. I believe that discussing this unconventional phenomenon would be very interesting and insightful.


 [1] Shen K, Ju Z, Tan X, et al. NaturalSpeech 2: Latent Diffusion Models are Natural and Zero-Shot Speech and Singing Synthesizers[C]//ICLR. 2024.

[2] Le, Matthew, et al. "Voicebox: Text-guided multilingual universal speech generation at scale." Advances in neural information processing systems 36 (2023): 14005-14034.

**Ethical Concerns:**

["NO or VERY MINOR ethics concerns only"]

**Final Justification:**

I appreciate the authors’ response in addressing my concerns, and I am willing to retain my current positive score.

**Limitations:**

yes

**Quality:**

2

**Strengths And Weaknesses:**

Strengths
1. Significant Compression Achievement: TaDiCodec achieves an unprecedented compression rate (6.25 Hz frame rate, 0.0875 kbps bitrate) while maintaining competitive reconstruction quality, representing a major advance in speech tokenization efficiency.

2. The paper includes extensive experiments across multiple languages, challenging test sets (articulatory, code-switching, cross-lingual), and both reconstruction and generation tasks.

3. The paper is written very clearly and is easy to read.

Weaknesses
1. Limited Application Scenarios: Although the title mentioned speech language modeling, the codec appears to be specifically designed for TTS tasks, rather than for speech language models that unify both speech understanding and generation. Due to its reliance on additional text information for decoding, the speech tokens are difficult to use for speech understanding tasks—especially those related to text, such as ASR. In contrast, the speech tokenizers used in models like Moshi and GLM-4-Voice are capable of handling both speech understanding and generation tasks.

2.  The comparison in Table 1 is not fair. Although the paper claims that the token rate has been compressed to an extremely low frame rate, the table is actually misleading. This is because codecs like EnCodec, DAC, and Mimi reconstruct audio directly from speech tokens, while some of these codecs require additional conditioning information for reconstruction. Such a comparison is therefore not fair. I believe that at least one more column should be added to indicate whether each codec requires extra information for reconstruction, such as: none, prompt speech, global features, or text condition, etc.

---

> ### Author Rebuttal · Authors · 2025-07-30
>
> Dear reviewer, we want to thank for your carefully reading and providing the constructive comments! Below we address the concerns mentioned in the review.
>
> `Weakness 1: Limited Application Scenarios... In contrast, the speech tokenizers used in models like Moshi and GLM-4-Voice are capable of handling both speech understanding and generation tasks.`
>
> In this work, we primarily focus on exploring the use of diffusion-based speech tokenizers for TTS and as output representations for speech language models. The motivation stems from the observation that, in TTS scenarios and most speech language models, the corresponding text of the target speech is typically available. For example, in TTS, the target text is always known, and in most end-to-end spoken dialogue systems, text and speech tokens are generated jointly. Under this setting, we aim to push the limits of compression in speech tokenization.
>
> We are actively investigating how diffusion speech tokenizers can be used for both speech generation and understanding. One promising direction is to inject textual information into the speech tokens during training, either through explicit ASR supervision or by enabling the encoder to generate an additional set of tokens aligned with the text—thereby eventually removing the decoder's dependence on textual input. This remains an exciting area for future work. In this paper, however, we focus on the text-aware decoder setting, where the decoder is conditioned on textual information during generation.
>
> `Weakness 2: The comparison in Table 1 is not fair... I believe that at least one more column should be added to indicate whether each codec requires extra information for reconstruction, such as: none, prompt speech, global features, or text condition, etc.`
>
> Thank you for your suggestion. We will revise the table in the paper accordingly and add an additional column to clarify the extra inputs required by each codec. We will also compute an adjusted token rate upper bound that accounts for these additional conditions, to provide a clearer and more accurate comparison.
>
> `Question 1: Regarding the diffusion decoder... Are these tokens encoding the residual information, maybe speech-text-global? Providing more analysis on this point would add valuable insights to the paper.`
>
> Thank you for the insightful question. Prior works such as [1, 2] predict mel-spectrograms or continuous latent features directly from text and prompts, but still require an additional duration predictor to explicitly align text and speech. Meanwhile, studies like [3, 4] have shown that it is possible to predict speech features without explicit alignment modeling. However, all of these approaches directly model complex continuous acoustic representations, which are often difficult to capture comprehensively the target distribution.
>
> Qualitatively, these models can be viewed as a **special case of TaDiCodec with a token rate of zero**. That is, they first predict text, encode the audio into "discrete tokens with sequence length of zero", then decode from text and "zero length tokens". It is evident that such approaches struggle to accurately reconstruct the original speech, especially in more expressive scenarios. Our demo page demonstrates that our model is capable of reconstructing challenging speech types such as singing, whispering, and speech with significant prosodic variation, with minimal timbre or pitch distortion.
>
> Returning to the TTS and language modeling setting, we argue that predicting TaDiCodec tokens and decoding them is more tractable than directly predicting high-dimensional continuous features. This is because the distribution over compressed discrete tokens is easier to model, and the tokens can be decoded into high-quality audio. Our TTS results, which outperform strong baselines such as F5-TTS, further support this claim.
>
> We also provide analyses to better understand what TaDiCodec learns. Since our codec achieves high-fidelity reconstruction, the learned tokens encode residual factors such as prosody, speaker information, and text-speech alignment. This is evident from our demo page, which showcases reconstructions of singing and other expressive speech styles. Furthermore, because our system is trained end-to-end without explicit disentanglement constraints, the tokens naturally capture some semantic information as well. This behavior reflects the tradeoff governed by the information bottleneck, which is influenced by factors such as codebook size and token rate. Our preliminary experiments show that with a codebook size of 16,384, the TTS model achieves excellent performance, suggesting that the alignment between text and speech tokens can be easily learned under this capacity. However, when the codebook size is reduced to 4,096, we observe that while the reconstruction quality remains high, TTS performance drops significantly. We attribute this to a loss of semantic content in the tokens, which makes it harder for the language model to learn the alignment between text and speech tokens.
>
> We also provide additional evaluations of our tokenizers on the DASB benchmark to show that TaDiCodec learns acoustic information. Due to time constraints, we select two representative tasks: emotion recognition and speaker verification. Following the official DASB protocol, we train models using our codec embeddings (after projection) and compare the performance against SpeechTokenizer as reported in the original DASB paper:
>
> | Model           | ER (Acc) | SV (EER) |
> |-----------------|----------|----------|
> | TaDiCodec       | 54.5     | 16.7     |
> | SpeechTokenizer | 49.1     | 20.4     |
>
> `Question 2: In Figure 3a, the WER of reconstruction is generally better than that of generation...  I believe that discussing this unconventional phenomenon would be very interesting and insightful.`
>
> Interesting question! This phenomenon has also been observed in some advanced TTS systems. We believe the underlying reason lies in certain biases introduced by the way WER is computed. For most test sets, paired speech and text are provided, and in theory, the ground-truth speech should yield a WER of zero. However, this is often not the case in practice, as ASR models learn to predict from an averaged distribution, and may not perfectly align with the real ground-truth audio.
>
> In contrast, the generated speech tokens are guided directly by the text, which in some cases may be more easily captured by the ASR model, resulting in lower WER.
>
> **A similar phenomenon is also observed between the reconstructed audio from TaDiCodec and the ground-truth speech**. For example, on the Chinese test set SeedTTS-zh, we find that TaDiCodec-reconstructed speech yields a better WER than the original ground-truth speech (0.94 vs. 1.254). One possible explanation is that TaDiCodec uses a text-aware decoder, which may lead to better alignment between the predicted speech and the corresponding text. From the perspective of an averaged ASR distribution, this alignment could make the generated speech more recognizable to the ASR model.
>
> Thanks again for your valuable comments, we would be grateful if we could hear your feedback regarding our answers. We would be happy to answer and discuss if you have further comments.
>
> Reference
>
> > [1] Shen K, Ju Z, Tan X, et al. NaturalSpeech 2: Latent Diffusion Models are Natural and Zero-Shot Speech and Singing Synthesizers[C]//ICLR. 2024.
>
> > [2] Le, Matthew, et al. "Voicebox: Text-guided multilingual universal speech generation at scale." Advances in neural information processing systems 36 (2023): 14005-14034.
>
> > [3] Eskimez S E, Wang X, Thakker M, et al. E2 tts: Embarrassingly easy fully non-autoregressive zero-shot tts[C]//2024 IEEE Spoken Language Technology Workshop (SLT). IEEE, 2024: 682-689.
>
> > [4] Chen Y, Niu Z, Ma Z, et al. F5-tts: A fairytaler that fakes fluent and faithful speech with flow matching[J]. arXiv preprint arXiv:2410.06885, 2024.

---

### Official Review · Reviewer_DCzz · 2025-07-03

**Clarity:** 3
**Significance:** 3
**Originality:** 3
**Rating:** 5
**Confidence:** 3

**Summary:**

This paper introduces TaDiCodec, a novel speech tokenizer designed to address several key limitations of existing methods. TaDiCodec uses a text-aware diffusion autoencoder to perform end-to-end optimization of both quantization and reconstruction, relying only on a diffusion loss. This approach eliminates the need for multi-stage training, auxiliary pre-trained models for semantic distillation, and complex adversarial objectives. The method achieves an extremely low token rate of 6.25 Hz with a single-layer codebook, while demonstrating strong performance in both speech reconstruction and downstream zero-shot text-to-speech (TTS) tasks.

**Questions:**

* Since the vocoder is not specified, it could be assumed that the RTF calculation excludes vocoder inference time. Please clarify this explicitly.
* While the contribution of this work and its significance is not diminished without this discussion, to broaden the work's impact on the wider community, it would be beneficial for the authors to discuss future directions for adapting TaDiCodec for streaming scenarios and its integration with real-time language models.

**Ethical Concerns:**

["NO or VERY MINOR ethics concerns only"]

**Limitations:**

yes

**Quality:**

3

**Strengths And Weaknesses:**

Strengths:
* The proposed method is well-motivated and intuitively designed. It effectively addresses several major challenges in speech tokenizer design, including achieving an ultra-low frame rate, removing the dependency on pre-trained semantic feature extractors, and simplifying the training pipeline to a single end-to-end objective based on diffusion loss.
* The paper provides a comprehensive and extensive comparison with numerous state-of-the-art audio tokenizers and downstream TTS systems.
* The experimental results are strong, demonstrating that TaDiCodec outperforms most baselines in both reconstruction quality and, more importantly, in downstream zero-shot TTS tasks.

Weaknesses:
* The paper lacks clarity on some key implementation details. For instance, the specific downsampling and upsampling methods used before and after quantization are not described. The paper states that it uses mel-spectrograms as the reconstruction target, but it does not specify which vocoder model was used to generate the final audio waveforms. This is a critical detail needed when evaluating it compared to other methods.
* While the current architecture, which relies on bidirectional attention and an iterative diffusion decoder, is primarily suited for audio compression and offline zero-shot TTS, this limits its applicability for streaming use cases where audio chunks must be generated sequentially before the full text token stream is available.

---

> ### Author Rebuttal · Authors · 2025-07-29
>
> Dear reviewer, we want to thank for your carefully reading and providing the constructive comments! Below we address the concerns mentioned in the review.
>
> `Weakness 1: The paper lacks clarity on some key implementation details. For instance, the specific downsampling and upsampling methods used before and after quantization are not described. The paper states that it uses mel-spectrograms as the reconstruction target, but it does not specify which vocoder model was used to generate the final audio waveforms. This is a critical detail needed when evaluating it compared to other methods.`
>
> Thank you for your suggestion! We are happy to provide more implementation details. Since our tokenizer operates at a frame rate of 6.25 Hz, while both the input and output mel-spectrograms are at 50 Hz, we perform downsampling before quantization and upsampling after quantization to match the temporal resolutions.
>
> To keep it simple, we use basic interpolation for both operations. Let the embedding before quantization be denoted as `z`, and the embedding after quantization as `z_q`. The following pseudocode illustrates this process using `torch.nn.functional.interpolate`:
> import torch.nn.functional as F
>
> ```python
> # Downsample from 50 Hz to 6.25 Hz before VQ
> z_downsampled = F.interpolate(z, scale_factor=1/8, mode="linear", align_corners=True)
> # Quantize
> z_q = vq(z_downsampled)
> # Upsample back to 50 Hz after VQ
> z_q_upsampled = F.interpolate(z_q, scale_factor=8, mode="linear", align_corners=True)
> ```
>
> We find this simple approach to be effective and sufficient for aligning the representations temporally across stages.
>
> For the vocoder, we use the Vocos [1] architecture with increased model capacity (around 120M parameters) and adopt the loss functions and GAN design from DACodec [2]. Unlike traditional vocoders that require upsampling, the Vocos decoder directly predicts the STFT spectrogram, which leads to higher inference efficiency compared to BigVGAN. Additionally, we find that it achieves strong reconstruction quality for both speech and singing.
>
> We will include this information in the appendix of the paper, and we commit to releasing our models and training code to further support transparency and reproducibility.
>
> `Weakness 2: While the current architecture, which relies on bidirectional attention and an iterative diffusion decoder, is primarily suited for audio compression and offline zero-shot TTS, this limits its applicability for streaming use cases where audio chunks must be generated sequentially before the full text token stream is available.`
>
> Thank you for the insightful comment! This is indeed one of the directions we plan to explore in future work. From an engineering perspective, even without modifying the model architecture, we can perform chunk-wise inference by generating a fixed number of text tokens, predicting the corresponding speech tokens, and decoding them with TaDiCodec, without waiting for all text tokens to be generated.
>
> A more promising direction is to design the decoder itself to be chunk-wise during training, supporting autoregressive diffusion over both text and speech tokens. Concretely, the current decoder input follows the pattern:
>
> ```
> [text tokens, speech tokens]
> ```
>
> We propose to restructure it as:
>
> ```
> [text_chunk_1, speech_chunk_1, ..., text_chunk_n, speech_chunk_n]
> ```
> This interleaved organization pattern of text and speech tokens has also been adopted in many speech LLMs, such as GLM4-Voice [3] and the streaming version of CosyVoice 2 [4].
>
> where the length of speech tokens in each chunk is set to a fixed ratio relative to the text tokens. Within each chunk, diffusion is used for generation, while causal attention is applied across chunks. This approach has been validated in prior works such as [5, 6].
>
> Our preliminary experiments also show that introducing a slight shift (or look-ahead) of approximately 1 second between speech tokens and the target mel-spectrogram improves chunk-wise streaming inference. Under a 1-second chunk size, the model can achieve real-time performance with minimal degradation in quality.
>
> `Question 1: Since the vocoder is not specified, it could be assumed that the RTF calculation excludes vocoder inference time. Please clarify this explicitly.`
>
> For the vocoder, we use the Vocos [1] architecture with increased model capacity (around 120M parameters) and adopt the loss functions and GAN design from DACodec [2]. Unlike traditional vocoders that require upsampling, the Vocos decoder directly predicts the STFT spectrogram, which leads to higher inference efficiency compared to BigVGAN. Additionally, we find that it achieves strong reconstruction quality for both speech and singing. It is worth noting that in the RTF calculation for TTS, we observe that the vocoder contributes minimally to the overall inference time across all baselines. The majority of the computational cost lies in the language model and the flow-matching decoder.
>
> `Question 2: While the contribution of this work and its significance is not diminished without this discussion, to broaden the work's impact on the wider community, it would be beneficial for the authors to discuss future directions for adapting TaDiCodec for streaming scenarios and its integration with real-time language models.`
>
> Thanks for your suggestion! We discuss this question in Weakness 2.
>
> Thanks again for your valuable comments, we would be grateful if we could hear your feedback regarding our answers. We would be happy to answer and discuss if you have further comments.
>
> Reference:
>
> > [1] Siuzdak H. Vocos: Closing the gap between time-domain and fourier-based neural vocoders for high-quality audio synthesis[J]. arXiv preprint arXiv:2306.00814, 2023.
>
> > [2] Kumar R, Seetharaman P, Luebs A, et al. High-fidelity audio compression with improved rvqgan[J]. Advances in Neural Information Processing Systems, 2023, 36: 27980-27993
>
> > [3] Zeng A, Du Z, Liu M, et al. Glm-4-voice: Towards intelligent and human-like end-to-end spoken chatbot[J]. arXiv preprint arXiv:2412.02612, 2024.
>
> > [4] Du Z, Wang Y, Chen Q, et al. Cosyvoice 2: Scalable streaming speech synthesis with large language models[J]. arXiv preprint arXiv:2412.10117, 2024.
>
> > [5] Liu Z, Wang S, Inoue S, et al. Autoregressive diffusion transformer for text-to-speech synthesis[J]. arXiv preprint arXiv:2406.05551, 2024.
>
> > [6] Ju Z, Yang D, Yu J, et al. MoonCast: High-quality zero-shot podcast generation[J]. arXiv preprint arXiv:2503.14345, 2025.

---

> > ### Comment · Reviewer_DCzz · 2025-08-07
> >
> > The authors have satisfactorily addressed all of my concerns. The clarifications regarding the interpolation method (W1) and the Vocos-based efficient vocoding (Q1) are clear. The discussion on future directions for streaming applications (W2 & Q2) was also insightful.
> >
> > I hope the authors will incorporate these clarifications into the final manuscript and proceed with open-sourcing their work as they commented. The paper is technically solid and makes a valuable contribution; therefore, I maintain my original score.

---

### Note · Authors · 2025-08-12

Dear ACs, SACs, and Reviewers,

We sincerely thank all reviewers again for the invaluable feedback and constructive suggestions. The discussions during the rebuttal period have greatly helped us improve our manuscript. We also appreciate the efforts of the ACs and SACs in organizing and coordinating the review process.

Below is a brief summary of the key points addressed during the rebuttal discussions:

1. **Understanding TaDiCodec Tokens and Compression Efficiency:** We discussed what we have learned about TaDiCodec tokens and why TaDiCodec achieves such high compression efficiency. Both reviewer rwbf (Q1) and rmsF (Q2) raised questions in this regard. Our analyses show that the learned tokens not only preserve residual acoustic factors such as prosody, speaker identity, and text-speech alignment, but also capture some semantic information due to the end-to-end training setup. We further demonstrate that the discrete token space, governed by codebook size and token rate, strikes an effective tradeoff between compression and information retention, enabling high-fidelity reconstruction and strong downstream performance.

2. **Additional Ablation Studies:** We provided further ablation experiments in the response to reviewer rmsF (W7) and WHX2 (Q2), including comparisons between **GAN vs. diffusion**, the effect of text quality and performance degradation when **text-speech misalignment**, and the effect of being **text-aware**.

3. **Broader Downstream Evaluations:** We reported TaDiCodec's performance on more downstream tasks beyond TTS. Specifically, we provide **additional evaluations of our tokenizers on the DASB benchmark** with two representative tasks: emotion recognition and speaker verification.

4. **Quantitative Analysis of Prosody Variations:** We included quantitative analysis discussing slight prosody shifts in TaDiCodec reconstructions due to stochasticity.

5. **Future Directions for Streaming Applications:** We proposed potential directions for adapting the current TaDiCodec framework to streaming applications. Reviewer DCzz noted that we have **satisfactorily addressed all concerns** and found our **discussion on future directions for streaming applications (W2 & Q2) insightful**.

Finally, we confirm that we will open-source all of our code (including inference and training) and model weights once the anonymity period ends, regardless of the final acceptance decision, in order to contribute to the community.

---

### Decision · Program_Chairs · 2025-09-17

**Decision:**

Accept (poster)

**Comment:**

This paper introduces TaDiCodec, a text-aware diffusion autoencoder for speech tokenization that addresses major limitations of existing methods by enabling end-to-end optimization of quantization and reconstruction without multi-stage training, pre-trained semantic models, or adversarial objectives. The method achieves a low token rate of 6.25 Hz and demonstrates strong performance in both speech reconstruction and downstream zero-shot TTS tasks. The paper is well-motivated, experimentally solid (even if it could be improved), and shows improvements over prior work.  Overall , the rebuttal has addressed most concerns and the paper is technically solid and impactful within its scope, and merits acceptance.